# Social state alters vision using three circuit mechanisms in *Drosophila*

Catherine E. Schretter[1], Tom Hindmarsh Sten[2,3], Nathan Klapoetke[1], Mei Shao[1], Aljoscha Nern[1], Marisa Dreher[1], Daniel Bushey[1], Alice A. Robie[1], Adam L. Taylor[1], Kristin Branson[1], Adriane Otopalik[1], Vanessa Ruta[2] & Gerald M. Rubin[1✉]

Animals are often bombarded with visual information and must prioritize specific visual features based on their current needs. The neuronal circuits that detect and relay visual features have been well studied[1–8]. Much less is known about how an animal adjusts its visual attention as its goals or environmental conditions change. During social behaviours, flies need to focus on nearby flies[9–11]. Here we study how the flow of visual information is altered when female *Drosophila* enter an aggressive state. From the connectome, we identify three state-dependent circuit motifs poised to modify the response of an aggressive female to fly-sized visual objects: convergence of excitatory inputs from neurons conveying select visual features and internal state; dendritic disinhibition of select visual feature detectors; and a switch that toggles between two visual feature detectors. Using cell-type-specific genetic tools, together with behavioural and neurophysiological analyses, we show that each of these circuit motifs is used during female aggression. We reveal that features of this same switch operate in male *Drosophila* during courtship pursuit, suggesting that disparate social behaviours may share circuit mechanisms. Our study provides a compelling example of using the connectome to infer circuit mechanisms that underlie dynamic processing of sensory signals.

Behavioural context is critical to how animals detect and interpret visual information. For example, when driving on a congested highway, it is imperative to focus on the movement of the car ahead while ignoring other environmental cues. Such focus, or feature-based attention[12], also occurs during certain behavioural states, including aggression, when it is important to be attuned to the movement of a competitor. Pioneering work across primates, rodents and invertebrates has shown the importance of neuronal populations tuned to specific features, including size, speed and colour[1–8]. Recordings in rodents suggest that visual processing regions receive input from non-sensory areas and circuits conveying information about behavioural state[13–18], allowing for modulation by context. However, the exact circuit architecture underlying state-dependent gating of visual attention remains unclear.

The fruit fly *Drosophila melanogaster* provides a powerful model for the mechanistic dissection of state-dependent visual processing due to its genetic accessibility, brain-wide connectome and complex behaviours. In flies, visual projection neurons (VPNs) compute the presence and general location of distinct visual features, such as looming or translating objects of varying sizes and speeds[19–21], and relay this information from the optic lobe to different target regions of the central brain[19,22–24]. These visual pathways appear to be highly stereotyped across sexes[19,25]. Selective activation of certain VPNs gives rise to robust behavioural outputs[19], some of which are context dependent[26–32]. For example, the same looming-responsive VPNs are involved in both landing and take-off, with the resultant behavioural output determined by octopaminergic modulation of downstream neurons[21,33,34].

During social behaviours, detection of conspecifics is critical[35–38]. Lobula columnar (LC) neurons, a major class of VPNs, project from the optic lobe to discrete brain glomeruli[19]. Three LC cell types, LC9, LC10a and LC11, that are tuned to fly-sized moving objects have been implicated in locomotor pursuit of conspecifics[9–11,19,23,24,39–42]. LC10a is one of the LC10-group cell types, or LC10s, each of which receive distinct inputs in the optic lobe. Previous work has shown that LC10a responses to fly-size objects increase substantially during male courtship pursuit[11]. Other LC10s do not appear to display this gain enhancement and their role in social pursuit is less well understood. Stimulation of courtship- and aggression-promoting P1 neurons[38] in males increases LC10a sensitivity to fly-sized objects, suggesting arousal-dependent modulation of visual processing[11]. Yet, in the absence of a male brain-wide circuit diagram, the circuit mechanisms underlying P1 modulation of LC10a remain unresolved. While P1 is only found in males, it represents a subset of the pC1 lineage that gives rise to pC1d and pC1e neurons in females[43–45], which also modulate social states. Recent work has shown that either simultaneous activation of pC1d and pC1e, or activation of the approximately 9 to 12 cells comprising the aIPg cell type can generate both acute aggressive behaviour and a persistent aggressive state in females[46–48]. Our previous research revealed that aIPg cells provide excitatory input to several LC10-group cell type targets, suggesting a role for aIPg cells in gating the flow of visual information[46]. Here we focus on female–female aggression

[1]Janelia Research Campus, Howard Hughes Medical Institute, Ashburn, VA, USA. [2]Laboratory of Neurophysiology and Behavior, The Rockefeller University, New York, NY, USA. [3]Present address: Department of Biology, Stanford University, Stanford, CA, USA. ✉e-mail: rubing@janelia.hhmi.org

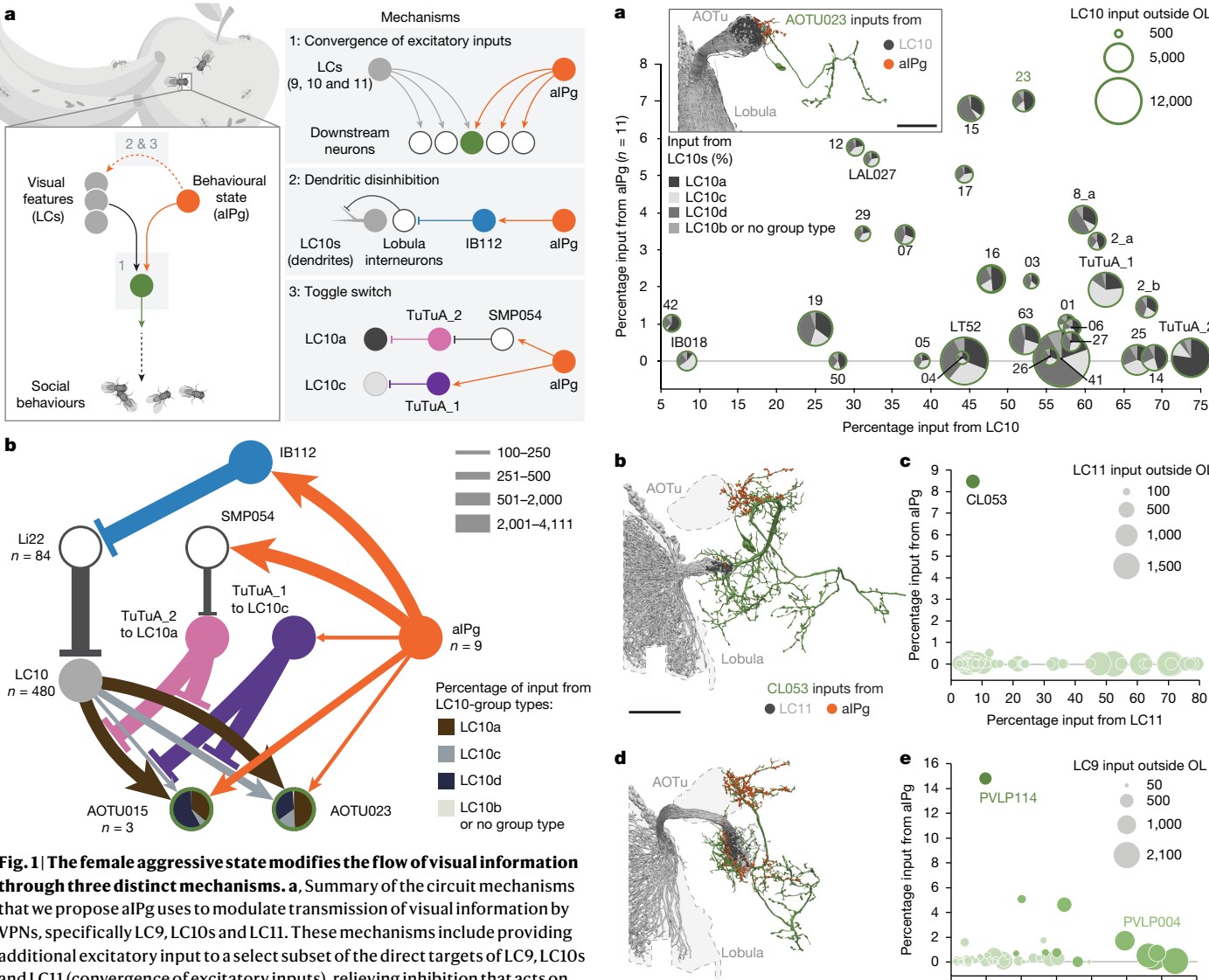

**Fig. 1 | The female aggressive state modifies the flow of visual information through three distinct mechanisms. a**, Summary of the circuit mechanisms that we propose aIPg uses to modulate transmission of visual information by VPNs, specifically LC9, LC10s and LC11. These mechanisms include providing additional excitatory input to a select subset of the direct targets of LC9, LC10s and LC11 (convergence of excitatory inputs), relieving inhibition that acts on the dendritic arbours of LC10s in the lobula (dendritic disinhibition) and simultaneously flipping a pair of switches that act on the axonal terminals of the LC10a and LC10c cell types to influence which of these two LC10s is active in signalling to downstream targets (toggle switch). Other than the LC9, LC10-group and LC11 targets discussed above, only three other neurons get both 1.5% or more of their input from aIPg and 5% or more of their input from an LC type. The arrows indicate putative excitatory connections (cholinergic) and the bar endings indicate putative inhibitory connections (GABAergic or glutamatergic). **b**, Overview of the circuit components for each mechanism in **a**. Activation of aIPg (1) provides additional excitatory input to downstream targets of LC10s, represented here by AOTU015 and AOTU023; (2) leads to disinhibition of inputs to the dendrites of LC10s through IB112; and (3) governs whether LC10a or LC10c is able to signal to their downstream targets by a toggle switch. This switch is operated by the TuTuA_1 and TuTuA_2 neurons, which provide inhibition to LC10c and LC10a, respectively. The line widths represent synaptic connections and are scaled according to the key. For cell types with more than one cell per brain hemisphere, the number of cells is indicated below the cell type. Additional details are provided in Extended Data Fig. 10.

and show that aIPg cells use three circuit mechanisms to modulate visual processing, which underlies this social behaviour (Fig. 1).

## Vision is critical for aggression

Multisensory cues are important for locating others and directing aggressive actions[37,46,49]. Previous behavioural evidence suggested

**Fig. 2 | Shared downstream targets of both aIPg and LC10s. a**, Shared downstream targets of aIPg and LC10-group cell types. Each target cell type is represented by a circle of which the diameter represents the total number of synapses that it receives from LC10-group cell types. AOTu cell types are indicated as numbers without the AOTU0 prefix. The proportion of those inputs coming from each LC10-group cell type is indicated in the pie chart. Postsynaptic sites from aIPg (orange) and LC10s (dark grey) on the outline of AOTU023 (dark green) are shown in the inset. **b**, CL053's morphology (dark green) is shown with the position of input synapses from aIPg (orange) and LC11 (dark grey). **c**, Shared downstream targets of both aIPg and LC11 neurons outside of the optic lobe (OL). Each target cell is represented by a light green circle of which the diameter indicates the total number of LC11 synapses that the cell receives and of which the position on the *y* axis represents the percentage of its inputs coming from aIPg and its position on the *x* axis represents the percentage coming from LC11. This graph shows LC11's top 51 targets outside the optic lobe, representing 74% of its synapses to other cell types outside the optic lobe. **d**, PVLP114's morphology (dark green) is shown with the position of input synapses from aIPg (orange) and LC9 (dark grey). **e**, Shared downstream targets of aIPg and LC9 neurons outside of the optic lobe. Each target cell is represented by a light green circle. The diameter of each circle indicates the total number of LC9 synapses that the cell receives, the position on the *y* axis represents the percentage of its inputs coming from aIPg and the position on the *x* axis represents the percentage coming from LC9. This graph shows LC9's top 54 targets outside the optic lobe representing 83% of its synapses to other cell types outside the optic lobe. For **a**, **b** and **d**, scale bars, 50 μm.

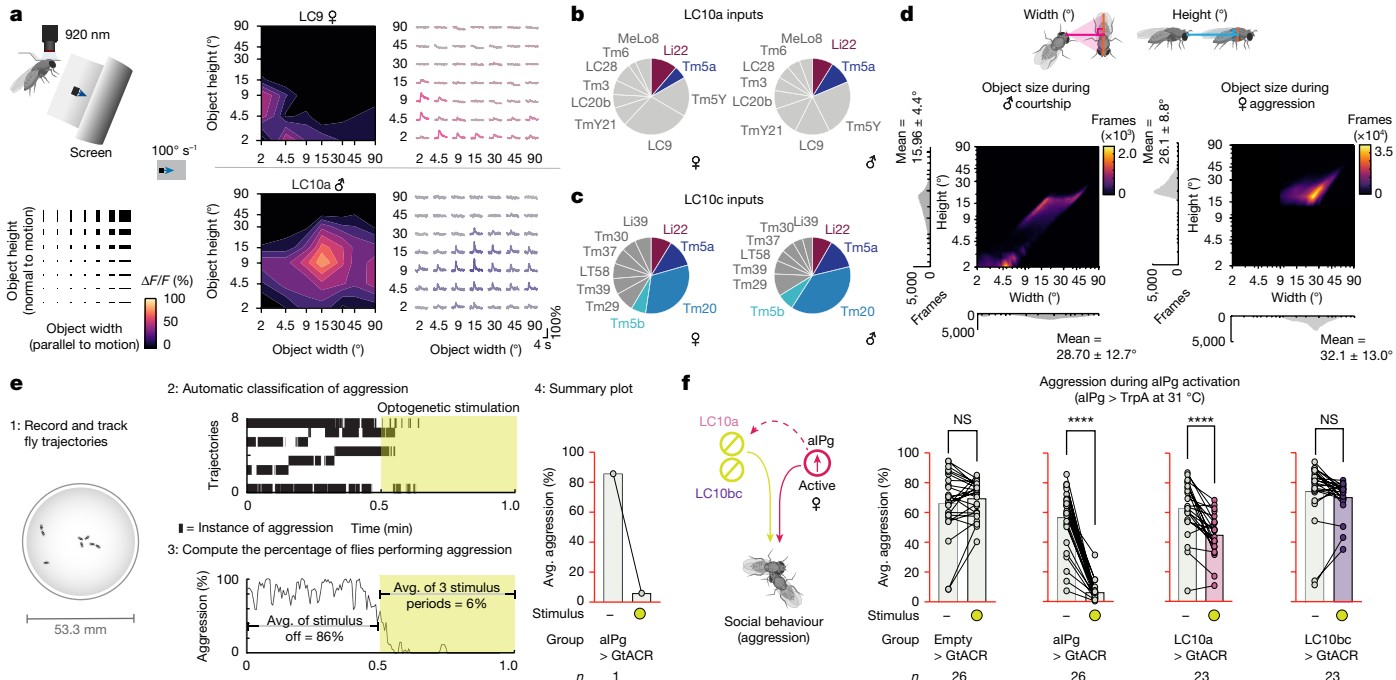

**Fig. 3 | LC10a is tuned to medium-sized moving objects and has a key role in female aggression. a**, Schematic of the experimental setup (top left) for presentation of moving dark rectangles of parameterized spatial dimensions (bottom left). Heat map representations and average traces (right) for individual LC9 and LC10a axons are shown across multiple animals. LC9: $n = 4$ female flies, $n = 4$ neurons; LC10a: $n = 5$ male flies, $n = 7$ neurons. **b,c**, The dendritic inputs input provided to LC10a (**b**) and LC10c (**c**) in the female and male by their top 10 input neurons as ranked in the male optic lobe. These inputs account for 31% and 40% of LC10a's inputs in the female and male, respectively, and 56% and 61% of LC10c's inputs in the female and male. **d**, Conspecific angular sizes experienced during male courtship and aIPg-induced female aggression. **e**, Schematic of the data analysis protocol,

representing one of the experiments in **f**. **f**, The average time spent performing aggressive behaviours before and during stimulus periods in which a 30 s continuous green (9 mW cm⁻²) light stimulus was delivered (green circles). The average over three stimulus periods is shown for display purposes. The time course and non-activating temperature controls are shown in Extended Data Fig. 3a–c. The average for the pre-stimulus period was calculated using the 15 s before the first stimulus period. All datapoints are shown to indicate the range, and the top edge of the bar represents the mean. Data were pooled from six independent biological replicates, which included separate parental crosses and were collected on different days. Statistical analysis was performed using nonparametric Wilcoxon matched-pairs signed-rank tests; asterisks indicate significance compared with 0; \*\*\*\**P* < 0.0001.

the importance of visual information in aIPg-induced aggression[46]. We confirmed this by eliminating the ability of females to receive visual information using a mutation in the *norpA* gene, which encodes a key component of the phototransduction pathway[50]. Activating aIPg in *norpA*-mutant females did not result in prolonged aggressive interactions, even after they made physical contact (Extended Data Fig. 1). These data emphasize the importance of visual cues in aggressive interactions elicited by aIPg activation.

## Shared targets of vision and state

To determine which visual pathways are modulated by aIPg, we performed a comprehensive analysis of the female connectome[25] and identified neurons receiving over 100 synaptic inputs from both aIPg and VPNs. We found that the VPNs participating in this circuit motif were a small subset (8 out of 44) of LC cell types. Moreover, each of these eight LC cell types is known to be responsive to small moving and/or looming objects[19,23,24,32]. This combination of aIPg and LC inputs may endow the downstream neurons with the ability to integrate an aggressive internal state with socially relevant visual information (Figs. 1 and 2). LC10s share the largest number of neuronal targets with aIPg. We identified 23 cell types that receive more than 25% of their input synapses from LC10-group cell types and about half of these also receive more than 2% of their input synapses from aIPg (Fig. 2a). Each of these shared outputs of aIPg and LC10s receives input from all LC10-group cell types, although in different proportions (Fig. 2a). Aside from LC10s, aIPg shares primarily one downstream target with LC11 (CL053) and two

downstream targets with LC9 (PVLP114 and PVLP004) (Fig. 2b–e). Only three other downstream neurons receive substantial input from both LC and aIPg neurons: PVLP120 receives 30% of its synaptic inputs from LC17, 19% from LC12 and 1.5% from aIPg; SMP312 receives 5.3% of its inputs from LC21 and 4.6% from aIPg; and PVLP006 receives 35% of its inputs from LC6, 11% from LC16 and 2.3% from aIPg. Thus, LCs and aIPg share a limited number of downstream targets, and these target cell types have the capacity to receive both excitatory visual and aIPg input during an aggressive encounter.

The vast majority of cell types that receive input from LC10s are interneurons in the anterior optic tubercle (AOTu), which then connect to descending pathways that drive motor action[41,51]. These interneurons may therefore control distinct facets of aggressive behaviour. Genetic reagents that enable us to target AOTu cell types, and perhaps combinations of cell types, as well as assays for subtle aspects of behaviour will be needed to further explore these parallel pathways.

Previous work characterizing visual responses of different LC populations using in vivo calcium imaging revealed that LC11 is selectively tuned to small moving objects (around 4.5° in angular width and height as subtended on the retina)[23,24,42]. We used the same experimental approach to examine visual feature selectivity of LC9 and LC10a. Similar to LC11, LC9 was preferentially tuned to smaller moving objects (around 4.5° in width and about 2° in height) in contrast to LC10a (around 15–30° in width and height) (Fig. 3a and Extended Data Fig. 2a,b). As perception of an object's size depends on its distance, such variations in size selectivity suggest differential activation of LC9 and LC10a when a female is far away from versus close to another fly,

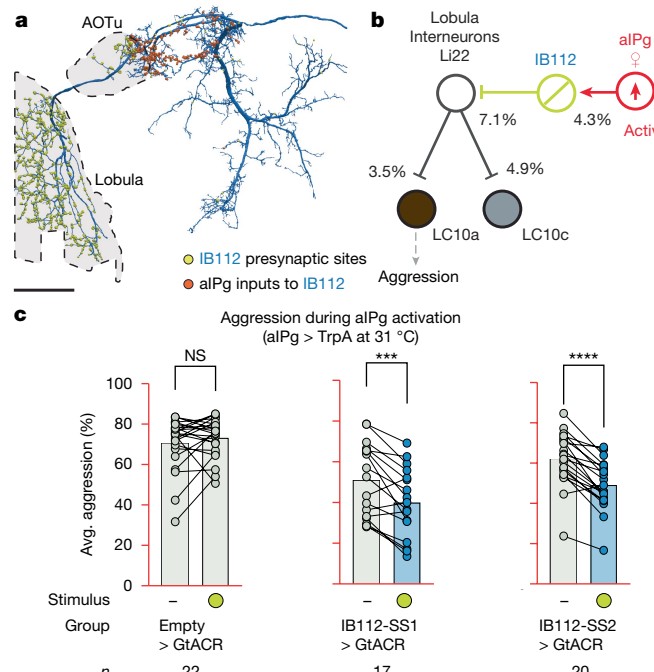

**a** AOTu

Lobula

**b** Lobula Interneurons Li22 · IB112 · aIPg ♀

IB112 presynaptic sites
aIPg inputs to IB112

7.1% · 4.3% · Active

3.5% · 4.9%

LC10a · LC10c

Aggression

**c**

Aggression during aIPg activation
(aIPg > TrpA at 31 °C)

| | NS | *** | **** |

Avg. aggression (%)

100
80
60
40
20
0

Stimulus: − ● · − ● · − ●

Group: Empty > GtACR · IB112-SS1 > GtACR · IB112-SS2 > GtACR

*n*: 22 · 17 · 20

**Fig. 4 | Polysynaptic connections from aIPg to the lobula shape aggressive behaviours. a**, Postsynaptic sites from aIPg (orange) and presynaptic sites of IB112 going to its downstream targets (yellow) in the lobula are shown on the neuronal outline of IB112 (dark blue). Scale bar, 50 μm. **b**, Overview of the strong connections between aIPg, IB112, Li22, LC10a and LC10c neurons. IB112 is aIPg's second strongest downstream target and receives 4.3% of its input from aIPg. Note that IB112 is named LoVC1 in the male optic lobe[53] and cL14 in FlyWire[54,59] datasets. The lobula interneuron Li22 (named Li01 in the FlyWire dataset[60]) is IB112's top downstream target, receiving over 40% of its output synapses (53% in males and 42% in females) in the optic lobe. Li22 cells receive 5.7% of their input from IB112 in the male optic lobe and 7.1% in the female optic lobe. Li22 provides 3.5% of LC10a, 1.1% of LC10b, 5.1% of LC10c and 5.9% of LC10d dendritic inputs in the male, and 3.5%, 0.8%, 4.9% and 5.3% in the female. **c**, The average time spent performing aggressive behaviours before and during stimulus periods in which a 30 s continuous green (9 mW cm⁻²) light stimulus was delivered. All datapoints are shown to indicate the range, and the top edge of the bar represents the mean. The time course and non-activating temperature controls are shown in Extended Data Fig. 5c–e. In the diagram in **b**, cell types inactivated with GtACR are circled in yellow and those activated with TrpA are circled in red. IB112 and the relevant lobula interneurons are glutamatergic and are presumed inhibitory (Extended Data Fig. 4h). Data were pooled from five biological replicates, which included separate parental crosses and were collected on different days. Statistical analysis was performed using nonparametric Wilcoxon matched-pairs signed-rank tests; asterisks indicate significance compared with 0; ***$P < 0.001$.

respectively. Moreover, LC9 displayed a prolonged calcium response to slow dark looming stimuli (Extended Data Fig. 2c). We found that, while the inputs to either LC10a or LC10c do not differ across sexes, inputs to these two cell types show little overlap, indicating that LC10a and LC10c respond to distinct visual features (Fig. 3b,c). For example, LC10c receives approximately 30% of its inputs from interneurons directly downstream of the R7 (Tm5a and Tm5b) and R8 (Tm20) photoreceptors that are critical for colour vision, whereas LC10a receives less than 4% of its input from such interneurons. Taken together, our results show that LC9, LC10a and LC11 are tuned to fly-sized moving objects and suggest that these LCs may be differentially activated over time, as a function of varying inter-fly distance, during aggressive encounters.

LC10a has been implicated in male courtship behaviour[10,11], yet the role of LC10a in female aggression has not been explored. To examine a potential role for LC10a in female social interactions, we first examined the visual object sizes and speeds experienced by female flies during aggressive encounters. During periods of aIPg-mediated aggression, female flies modulate their distance with respect to the nearest fly such that the angular size of that fly remains 32.1 ± 13.0° in width (mean ± s.d.) and 26.1 ± 8.8° in height, which corresponds to the flies being less than a body length apart (Fig. 3d). Such stimuli are within the preferred ranges of object size and speed of LC10a neurons (Fig. 3a and Extended Data Fig. 2d,f), consistent with the notion that LC10a could have a role during female aggression. The preferred range of stimuli for LC10a is also within the range of object sizes (28.70 ± 12.7° in width and 15.96 ± 4.4° in height) that courting males fixate on during courtship (Fig. 3d and Extended Data Fig. 2e,f), similar to previous work[10].

To assess the involvement of specific cell types in aIPg-mediated aggression, we recorded, tracked, classified and analysed the aggressive behaviour of groups of female flies in an arena as diagrammed in Fig. 3e. We quantified the average percentage of trajectories in which flies were performing aggression in each such experiment, which we represented as a single point in our graphs, and performed approximately 20 experiments per condition (Fig. 3e). For different LC types, we performed epistasis experiments in which we thermogenetically activated aIPg cells while optogenetically silencing an individual LC type (Fig. 3f). Optogenetic inactivation of LC10a, but not LC10bc, LC9 or LC11, resulted in a sustained decrease in aggressive behaviours, including individual component features such as touch (Fig. 3f and Extended Data Fig. 3a–g). When LC10a was silenced, the distance between flies increased, and the angular size subtended on the retina by the nearest fly as well as the number of flies within two body lengths correspondingly decreased (Extended Data Fig. 3c,d). Collectively, these results suggest that LC10a has a similar role in the visuomotor tracking of social targets across sexes.

## Broad disinhibition of visual inputs

Our connectomic analyses suggest a second mechanism by which aIPg enhances information flow from LC10s (Fig. 1). IB112, aIPg's second highest target based on synapse count, has nearly 90% of its synaptic output in the lobula (Fig. 4a,b), suggesting a role in modulating visual inputs to the brain. We generated two cell-type-specific driver lines for IB112 and confirmed that this cell type is glutamatergic, suggesting that it is probably inhibitory[52] (Extended Data Fig. 4). Whole-cell patch-clamp recordings and calcium imaging of IB112 during aIPg stimulation in female brain explants confirmed direct functional connectivity between aIPg and IB112 (Extended Data Fig. 5a,b). Moreover, inactivation of IB112 resulted in a prolonged decrease in aggression, including touch and other component behavioural features (Fig. 4c and Extended Data Fig. 5c–e), confirming the importance of IB112 outputs in the lobula.

We used the recently completed optic lobe connectome in males[53] to examine the synaptic outputs of IB112 in this dataset (Fig. 4b) and then verified the presence of these connections in the less extensively annotated female optic lobe connectome[54] (additional details in Methods). Within the male and female lobula, IB112 sends over 40% of its synapses to the lobula-intrinsic interneuron Li22, providing over 5% of Li22's input. No other downstream cell type receives more that 1.5% of their input from IB112 in either sex. Li22 in turn provides a large number of synapses to the dendrites of LC10-group cell types in females and males (additional details are shown in Fig. 4b). Li22 is predicted to be glutamatergic, implying that connections from this cell type are probably inhibitory[52].

IB112 is an example of a centrifugal neuron, defined as such by having most of its inputs within the central brain and its outputs in the optic lobes[53,54]. Analysis of the connectome has identified more than 70 centrifugal cell types[53]. While two previous examples of centrifugal neurons have been predicted to use neurotransmitters, such as octopamine, to modify behaviour[55,56], their circuit-level functions are

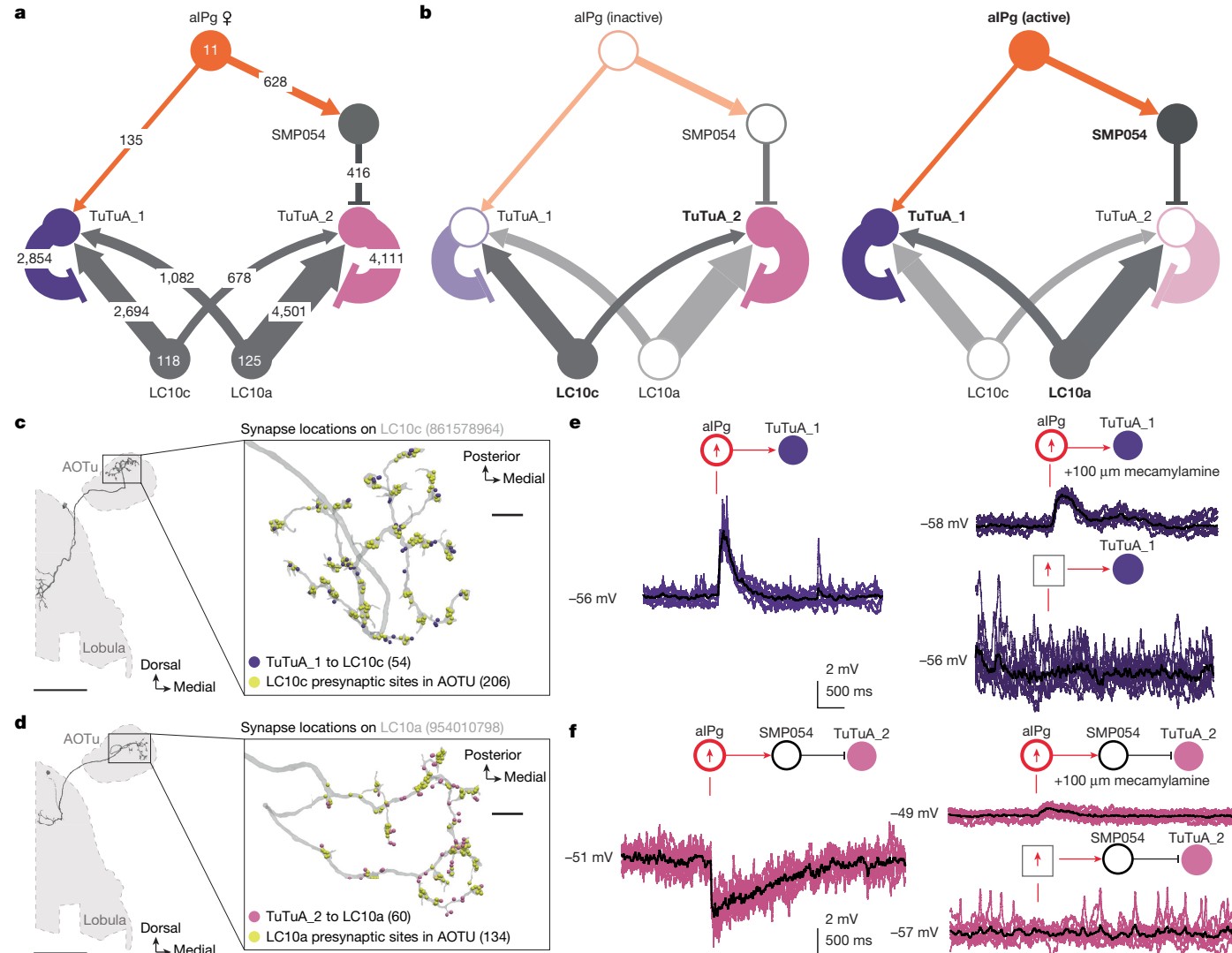

**Fig. 5 | aIPg selectively amplifies LC10a, while dampening LC10c transmission through TuTuA neurons. a**, The connectivity between aIPg, TuTuAs, LC10a and LC10c in one brain hemisphere. Synapse numbers are indicated on the arrows. TuTuAs connect to specific LC10s: 98% of TuTuA_1's synapses onto LC10s go to LC10c and 98% of TuTuA_2's synapses go to LC10a. TuTuA_1 and TuTuA_2 make <1% of their synapses onto LC10b, LC10d, LC10e or LC10f. The arrows indicate putative excitatory connections (cholinergic) and the bar endings indicate putative inhibitory connections (SMP054, GABAergic; TuTuA_1 and TuTuA_2, glutamatergic). **b**, Predicted outcomes for circuit dynamics based on aIPg activity. Cells and connections with higher predicted activity are displayed in bold font and dark colours. **c,d**, Synapses between TuTuA_1, TuTuA_2, LC10a and LC10c on representative skeletons for LC10c (**c**) and LC10a (**d**). Note how inhibitory synapses from the TuTuA neurons are interspersed with LC10s' output synapses. **e**, Excitatory responses recorded

from TuTuA_1 (*n* = 16 cells) in female brain explants in response to a 2 ms stimulation of aIPg neurons (72C11-LexA > Chrimson). The excitation was largely abolished by mecamylamine, an *n*-AchR blocker. No evoked response was recorded when stimulating Chrimson in the absence of a LexA driver (*n* = 5; bottom right). Individual trials are shown in purple (*n* = 8 trials from 1 cell) and the mean is shown in black. **f**, Inhibitory responses recorded from TuTuA_2 (*n* = 16 cells) to a 2 ms stimulation of aIPg neurons. The inhibition was completely removed by mecamylamine. No evoked response was recorded when stimulating Chrimson in the absence of a LexA driver (*n* = 5; bottom right). Individual trials are shown in pink (*n* = 8 trials from 1 cell) and the mean is shown in black. In the diagrams above the traces, cell types activated with LexAop-Chrimson are circled in red and those recorded from are shown in in purple or pink depending on the TuTuA cell type. For **c** and **d**, scale bars, 50 μm, and 5 μm in the inset images.

unclear. Thus, IB112 provides an example of this important class of inputs to the optic lobe for which not only a behavioural role but also the relevant direct synaptic inputs in the central brain and outputs in the optic lobe have been identified.

## A toggle switch

Our previous work identified the TuTuA neurons as potential targets of aIPg that could regulate LC10s signalling in the AOTu[46]. Further analysis of the female connectome revealed that there are two TuTuA cell types,

referred to as TuTuA_1 and TuTuA_2, each represented by a single glutamatergic cell per brain hemisphere. We found that the TuTuAs have distinct patterns of connectivity that suggest they differentially gate LC10a and LC10c (Fig. 5a). Given the connectivity and predicted neurotransmitters in this circuit, our simplest interpretation is as follows: when active, aIPg provides excitatory input to TuTuA_1, which forms connections interspersed with LC10c's output synapses and inhibits its ability to transmit information (Fig. 5b,c). Moreover, aIPg indirectly targets TuTuA_2 through the GABAergic interneuron SMP054. TuTuA_2 forms inhibitory connections with LC10a that are interspersed with its

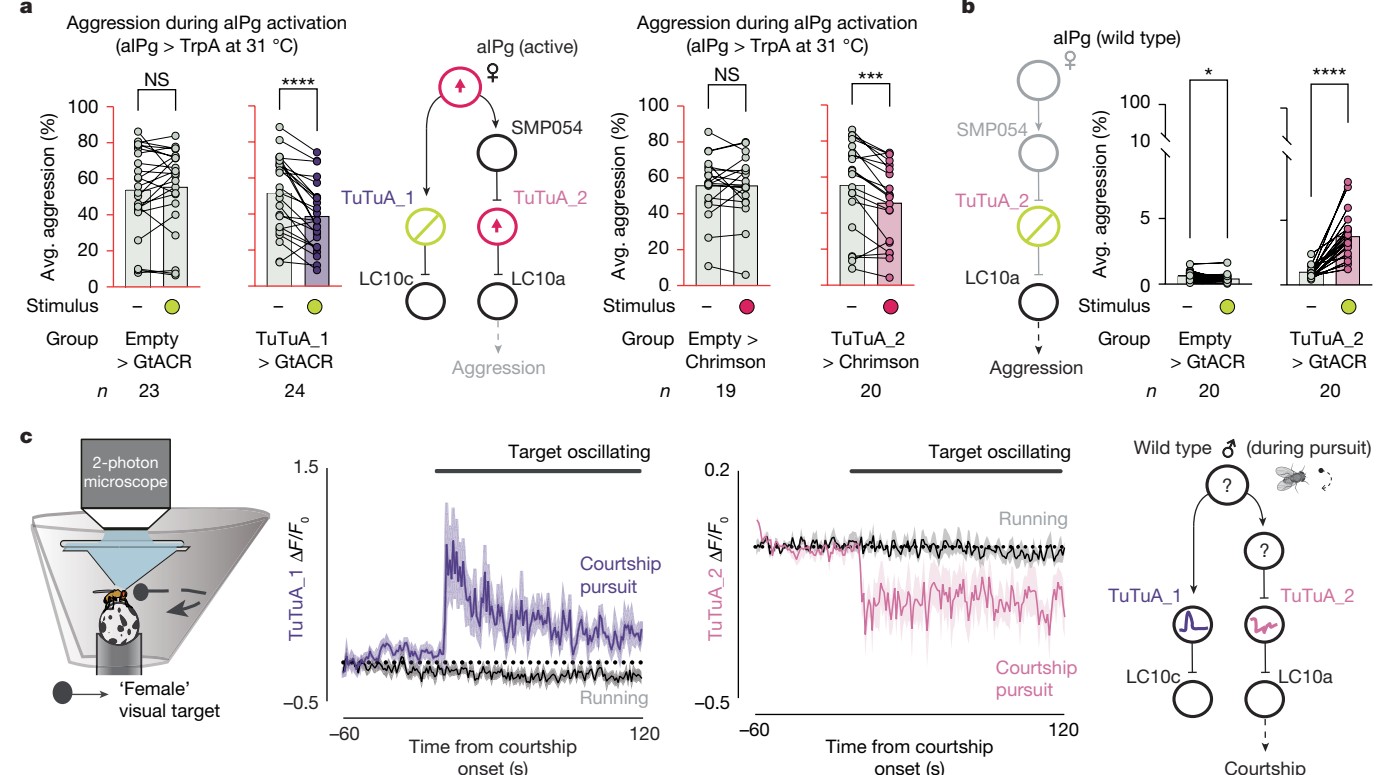

**Fig. 6 | Selective modulation of TuTuA_1 and TuTuA_2 shapes female aggression and male courtship. a,b,** The average time spent performing aggressive behaviours before and during periods in which a 30 s continuous green (9 mW cm$^{-2}$) or red (3 mW cm$^{-2}$) light stimulus was delivered. The time course and non-activating temperature controls are shown in Extended Data Fig. 8a–f. The 10 s (**a** (left)) or 30 s (**a** (right) and **b**) before each stimulus period was compared to average aggression during the same time intervals across all three stimulus periods. The empty > GtACR control in **b** displayed a small change in aggression after stimulation, but in the opposing direction to the experimental group. Experiments were performed as shown in Fig. 3e. All datapoints are shown to indicate the range and the top edge of the bar represents the mean. Cell types inactivated with GtACR are circled in yellow and those activated with either TrpA or Chrimson are circled in red. Data were pooled from six (**a** (left)) and three (**a** (right) and **b**) biological replicates. **c,** Schematics of the visual virtual reality preparation for male courtship (left, adapted from ref. 11) and circuit activity during male courtship pursuit (right) are shown. The identity of cell types indicated by question marks are not known. Responses of TuTuAs (centre, average $\Delta F/F_0$) to a visual target during periods of courtship pursuit (purple or pink) or during general locomotion (black). The mean is represented as a solid line, and the shaded bars represent standard error between experiments (TuTuA_1-SS1: $n = 4$ (courting), $n = 9$ (running); TuTuA_2-SS1: $n = 5$ (courting), $n = 6$ (running) flies). The black line above indicates when the visual target was oscillating. Statistical analysis was performed using nonparametric Wilcoxon matched-pairs signed-rank tests; asterisks indicate significance compared with 0; *$P < 0.05$.

output synapses. This disynaptic sequence of inhibitory connections serves to disinhibit LC10a transmission (Fig. 5b,d). In this simple view, aIPg activation has the potential to oppositely modulate LC10a and LC10c and their respective downstream circuits. Should aIPg have only two states—active and silent—the TuTuA neurons would act as a pair of opponent switches that flip from LC10c to LC10a transmission when aIPg becomes active.

TuTuA_1 and TuTuA_2 are predicted to have opposite effects on the activity of LC10a and LC10c (Fig. 5b). Excitatory feedback from LC10a and LC10c to TuTuA_1 and TuTuA_2, respectively, probably reinforces this opponency (Fig. 5b). Specifically, the connectome revealed that a major downstream target of LC10a is TuTuA_1, the same TuTuA that inhibits LC10c. Thus, when LC10a is active it serves to reinforce the suppression of LC10c activity. Analogous to LC10a's connections to TuTuA_1, a major LC10c target is TuTuA_2. This complementary indirect pathway serves to suppress LC10a activity when LC10c is active.

In the simplest interpretation of the circuit, TuTuA_1 and TuTuA_2 would function as a reciprocal on/off switch for LC10a and LC10c. However, we recognize that our interpretation of the mechanisms suggested by the circuit diagram are most likely incomplete, and there are features of this circuit that cannot be fully understood from the connectome alone. For example, both TuTuAs provide inhibition on distinct LC10-group cell types and receive substantial excitatory

feedback from those same LC10s. It is possible that this circuit motif provides inhibitory feedback to regulate the gain of TuTuA output[56–58]. Furthermore, how the neurons of this circuit integrate synaptic connections over time and at a subcellular level remain unclear and would require pointed neurophysiological interrogation beyond the current technical capacities.

We were able to test many of the predictions from the circuit diagram by generating GAL4 driver lines specific for each of the TuTuAs (Extended Data Fig. 6), and using these genetic reagents in functional assays. Electrophysiology and calcium imaging during aIPg stimulation confirmed several predicted connections: direct aIPg excitatory connections to TuTuA_1; indirect inhibitory connections to TuTuA_2; and excitatory connections of LC10a to both TuTuA_1 and TuTuA_2 (Fig. 5e,f and Extended Data Fig. 7a–e). pC1d and pC1e neurons, which were previously shown to be upstream of aIPg in the female aggression circuit[46], were also found to provide indirect inhibitory inputs to TuTuA_2 through SMP054 (Extended Data Fig. 7f–k).

To test the behavioural effects of this circuit architecture, we acutely silenced TuTuA_1 while activating aIPg. As expected, we found that optogenetic inhibition of TuTuA_1 activity during chronic aIPg activation transiently decreased aggression (Fig. 6a and Extended Data Fig. 8a,e). Next, we optogenetically activated TuTuA_2 during chronic aIPg thermogenetic activation and found that it significantly reduced

female aggressive behaviour throughout the duration of the stimulus (Fig. 6a and Extended Data Fig. 8b–e). Furthermore, TuTuA_2 optogenetic inactivation increased aggression behaviour in the absence of aIPg activation (Fig. 6b and Extended Data Fig. 8f). Taken together, these results provide strong evidence in support of a toggle switch mechanism whereby aIPg activation shifts the relative gain of the LC10a and LC10c visual pathways.

Note that the targets of LC10s within the AOTu are regulated in two ways by aIPg. First, aIPg provides substantial direct input to about half of these AOTu interneurons (Fig. 2a). Second, aIPg activation is predicted to produce a global shift in the visual input those neurons receive—even those that do not receive direct aIPg inputs—by gating whether LC10a or LC10c can effectively signal to them (Figs. 1b and 5). Thus, aIPg is primed to regulate the flow of visual information through all visual AOTu interneurons using either one or both of these distinct circuit mechanisms.

## The TuTuA switch functions in males

Previous work has demonstrated that P1 neurons, directly or indirectly, increase the gain of LC10a activity, but not LC10c activity, in the AOTu during courtship pursuit[11]. We previously suggested that a common mechanism involving the TuTuA neurons might underlie state-dependent gating in females and males[46]. Consistent with this suggestion, we show that TuTuA neurons exhibit similar morphology across sexes and share aspects of their baseline physiological properties (Extended Data Figs. 6p–s and 9). We next examined TuTuA activity in tethered males as they spontaneously initiated courtship pursuit of a 'fictive female' represented as a high contrast dot that moves at a constant angular velocity across the male's visual field[11] (Fig. 6c). While both TuTuA neurons were insensitive to the visual profile of the fictive female when males viewed it passively, the onset of courtship marked a striking change in their calcium activity: the activity of TuTuA_1 increased whereas that of TuTuA_2 decreased (Fig. 6c). These results are consistent with our circuit model in females, supporting the notion that the same TuTuA-mediated switch may also be used in males to gate visual processing during social interactions.

## Concluding remarks

Animals gate visual information in a context-dependent manner. Using the connectome as a guide, our work provides a detailed circuit-level understanding of how this can be accomplished. We found three distinct mechanisms, under coordinated control by a single cell type conveying internal state, that selectively amplify visual information critical for social interactions (Fig. 1b). This cell type, aIPg, appears to be largely dedicated to this task with its six most highly connected synaptic targets contributing to the gating of the visuo-motor circuits we described. These circuits are engaged by aIPg and have the potential to regulate distinct motor programs (Extended Data Fig. 10). The presence of these multiple circuit mechanisms endows the system with more degrees of freedom and flexibility in regulation of attention toward different visual features. It is difficult to imagine how we could have efficiently discovered these circuit mechanisms without the connectome. aIPg is the primary sexually dimorphic neuron in the circuits we described, while other circuit components—the LC neurons, lobula interneurons, IB112 and TuTuA neurons—are present in males and females with indistinguishable morphologies. Our data on TuTuA activity in males during tracking of a fictive female stimulus combined with analysis of the connectome of the male optic lobe suggest that the same toggle switch operates across sexes. Together, our results illustrate how a single node that differs across sexes could regulate multiple shared sensorimotor circuits. Our observations imply that these same circuit mechanisms and cell types will have a role in a range of social behaviours in which a fly must focus its attention on other nearby flies.

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

## Methods

### Fly stocks

All experiments used mated female flies unless otherwise stated. Flies were reared on standard cornmeal molasses food at 25 °C and 50% humidity. For optogenetic activation experiments, flies were reared in the dark on standard food supplemented with retinal (Sigma-Aldrich) unless otherwise specified, 0.2 mM all trans-retinal before eclosion and 0.4 mM all trans-retinal after eclosion. Hemidriver lines were created using gateway cloning as previously described[61]. Stable split GAL4 lines used in this study were constructed as described previously[61] and hemidrivers used are described in Supplementary Table 1. iLexA stands for improved LexA, in which additional activating domains were added the transcription activator to enhance expression[62]. For the aIPg-iLexA line, the pBPnlsLexA::p65::GADUw vector was used for cloning as described previously[48]). Original confocal image data of GAL4 lines are available at https://www.janelia.org/split-GAL4. Genotypes used in each figure are listed in Supplementary Tables 1, 3–5, 7, 9–11, 13, 15, 17, 18, 20–22, 24, 26–30, 32–34, 36 and 37.

### Thermogenetic and optogenetic activation behavioural experiments

Groups of 5–8 group-housed mated female flies (7–10 days after eclosion) were video recorded at 60% relative humidity in a 53.3 mm × 3.5 mm circular arena. This assay and automated analysis pipeline was used as described[63]. All non-thermogenetic (TrpA) experiments were performed at 24 °C, while thermogenetic experiments were performed at 22 °C for non-activating controls and 31 °C for activation experiments. All tests were conducted under visible light conditions at zeitgeber time 0 to zeitgeber time 4 unless otherwise stated. Flies were loaded into the arena using an aspirator. For activation of neurons expressing Chrimson, the arena was illuminated as specified in the figure legends using constant uniform illumination with 660 nm LEDs. For inactivation of neurons expressing GtACR, we used constant uniform illumination with 525 nm LEDs. All of the trials were performed under white-light illumination from above. Videos were recorded from above using a camera (USB 3.1 Blackfly S, Monochrome Camera; Point Grey) with an 800 nm long-pass filter (B+W filter; Schneider Optics) at 170 frames per second and 1,024 × 1,024 pixel resolution.

### Behavioural classification and analysis

For each trial, flies were acclimatized to the arena for 30 s before the delivery of six sets of constant stimuli each lasting 30 s with 30 s in between each stimulus. For all experiments, only the lowest stimulus intensity in which an effect was found is depicted and was analysed. Unless otherwise stated, the pre-stimulus average was calculated from the three periods before the stimulus periods used for analysis. In Fig. 3f, the previous stimuli appeared to alter behaviour during successive stimulus-off periods; therefore, only the first 15 s of the first pre-stimulus period was used for comparison. The videos were processed using the automated pipeline described previously[63]. In brief, flies were tracked using Caltech FlyTracker followed by automated classification of behaviour with JAABA classifiers[64]. Novel classifiers for touch and aggression were created based on previous definitions[46,65]. We validated the performance of these classifiers against manually labelled ground-truth data using videos that were not part of the training dataset (framewise performance is shown in Supplementary Table 2). As aggression is inherently the interaction of more than one fly, we cannot disassociate the behaviour of one fly from that of the others. Aggression is therefore computed on a per-arena basis rather than a per-fly basis. The percentage aggression is the sum of the aggression scores for all the trajectories in which flies were performing aggression divided by the total number of flies in the arena times 100. For figures displaying behavioural time courses, the mean of 0.35 s (60-frame)

bins is shown. The angle occluded by nearest fly was calculated using the anglesub perframe feature[65] and was computed by finding the two lines tangent to the fit ellipse that intersect at the nose-point of this fly, and measuring the angle between them.

For dyadic courtship assays ($n = 13$), the courtship start frame was manually identified based on first-instance inter-fly distance of <3 mm and fixation angle of <|20°| lasting for >1 min. Courtship end was defined as first copulation frame or end of video acquisition (30 min).

For calculation of visual features experienced during aggressive and courtship interactions, the angular position, velocity, height and width were calculated on a frame-by-frame basis using a custom MATLAB (MathWorks) script whereby, for each frame, the coordinates and orientations of subject fly and nearest conspecific, or target fly, were translated and rotated such that the subject was situated at the origin facing zero degrees. In this new basis, the target fly's angular position ($\theta$) and velocity ($\varphi$) with respect to the subject fly's visual field were calculated as $\theta = \tan^{-1}\left(\frac{y}{x}\right)$ and $\varphi = \frac{d\theta}{dt}$, respectively. To approximate the angular size and expansion of the target fly in the subject's visual field, an ellipse was fit to the major and minor axes of the target fly in this new basis. The target fly's angular width was approximated as the length of the cross-section of this ellipse that lies perpendicular to the Euclidean line between the anterior-most point of the subject fly and target fly centroid. Thus, for each frame, the equation for the target fly's angular width, $w$, is $w = 2\left(\tan^{-1}\frac{R}{d}\right)$, where $R$ is half the real cross-sectional length of the ellipse (in mm) and $d$ is the Euclidean distance (in mm) between the anterior-most point of the subject fly and target fly centroid. The target fly's angular height was approximated in a similar manner; however the target fly's real height was fixed at 1 mm (which a reasonable estimation of a female fly's height), so the equation for angular height, $h$, was simply $h = 2\left(\tan^{-1}\frac{0.5\,\text{mm}}{d}\right)$, where $d$ is the Euclidean distance (in mm) between the anterior-most point of the subject fly and target fly centroid. Female aggression frames were defined using the JAABA aggression classifier and calculated from 79 trajectories.

### Fly preparation for calcium imaging

For Fig. 3a and Extended Data Fig. 2a–d, experiments were performed similarly to those described previously[24] with a similar preparation to that described in a previous study[66]. Notably, the fly's head was positioned and glued to the fly holder such that the eye's equator faced the middle of the visual projection screen. The proboscis remained intact but was glued in position, and a dissection needle was used to remove the cuticle and sever muscle 16.

For Fig. 6c, experiments were performed similarly to as described previously[11]. In brief, the flies were anaesthetized on $CO_2$ and tethered to a custom-milled plate. The flies were held in place by a string across the neck and fixed to the holder by both eyes and the back of the thorax using UV-curable glue. The proboscis was also glued to the mouthparts to minimize brain motion. Flies were left to recover in a warm, humidified chamber (25 °C, 50–70% humidity) in the dark for 1–4 h. The cuticle was subsequently dissected from the top of the head and flies were transferred to an air-supported foam ball.

### Two-photon calcium imaging

In Fig. 3a and Extended Data Fig. 2a–d, calcium imaging experiments were performed with male or female flies (LC10a and LC9 experiments respectively) 5–10 days after eclosion, maintained under standard conditions (21.8 °C, 55% humidity, 16 h–8 h light–dark, standard cornmeal/molasses food). A full list of fly genotypes used in the calcium imaging experiments is provided in the Supplementary Information. The imaging setup is identical to the previously described two-photon microscope (Thorlabs) setup[24]. In brief, we used a Ti:Sapphire femtosecond laser (Spectra-Physics Mai Tai eHP DS) tuned to 920 nm and delivering <20 mW power at the sample. Fluorescence signals were collected using a ×16 water-immersion objective (Nikon CFI75, NA 0.8) with a band-pass filter (Semrock 503/40 nm) in front of the photomultiplier

tube (Hamamatsu GaAsP H10770PB-40 SEL). Oxygenated saline was circulated throughout. Imaging volumes were acquired at 5.6 Hz or higher. Visual stimuli were delivered to the fly's right eye and all imaging was from the right side of the brain. The stimuli were presented on a screen that subtended approximately 90° by 90° of the fly's field of view with a green (532 nm) projector setup as previously described[24].

In Fig. 6c, calcium imaging experiments were performed with TuTuA_1 and TuTuA_2 male flies 3–7 days after eclosion, maintained under standard conditions (25 °C, 65% humidity, 12 h–12 h light–dark, standard Würzburg food). The imaging preparation for tethered courtship was identical to that previously described[11]. In brief, male flies rested and walked on a small 6.35 mm diameter ball, which was shaped from foam and manually painted with uneven black spots using a Sharpie. The foam ball was held by a custom-milled aluminium base and floated by air supplied at ~0.8 l min⁻¹ such that the ball could move smoothly. The ball was illuminated by infrared LED flood lights, and imaged with a Point Grey FLIR Firefly camera using a mirror. The ball was surrounded by a 270° conical screen with a large diameter of ~220 mm, a small diameter of ~40 mm and a height of ~60 mm. As males walked on the foam ball, all three rotational axes of the ball were read out by the FicTrac2.0 software[67] at 60 Hz in real-time. The visual stimulus was projected around the male from a DLP 3010 Light Control Evaluation Module (Texas Instruments) through a first-surface mirror below the fly. The red and green LEDs in the projector were turned off, leaving only the blue LEDs to minimize interference with GCaMP emissions.

Visual stimuli were generated in the MATLAB-based ViRMEn software[68] and projected onto the screen using custom perspective transformation functions. The net visual refresh rate of the visual stimulus ranged from 47.6 Hz to 58.9 Hz. Each trial was initiated by the presentation of a stationary visual target for 60 s to examine the animal's baseline locomotion, after which the visual target began to oscillate. The visual target oscillated in a 107° arc around the animal with a constant angular velocity of approximately 75° s⁻¹, but the angular size of the dot was continuously altered to mimic the dynamics of a natural female during courtship. The angular size was altered by changing the distance between the male and the target in the ViRMEn world. The distance between the male and the target was taken from the inter-fly distance in a courting pair over the course of two minutes of courtship and, at each frame, the angular position of the target was scaled by this inter-fly distance to give rise to a more dynamic female path. Angular sizes ranged between around 8 and 50°, with the average size being 22.5°. Each stimulus frame was therefore unique for 2 min of time, and subsequently repeated until the end of the trial when it intersected its original position. Each trial lasted 10 min.

Male imaging experiments were performed using the Ultima Investigator or Ultima Investigator Plus two-photon laser-scanning microscope (Bruker Nanosystems) with a Chameleon Ultra II Ti:Sapphire laser. All of the samples were excited at a wavelength of 920 nm, and emitted fluorescence was detected with a GaAsP photodiode detector (Hamamatsu). All images were acquired using a ×40 Olympus water-immersion objective with 0.8 NA. All images were collected using PrairieView Software (v.5.5 or 5.7) at a resolution of 512 px × 512 px.

Courtship and running was classified based on the fidelity and vigour of a male's pursuit of the visual target, as described previously[11].

In Extended Data Figs. 5b and 7a–e, ex vivo calcium imaging experiments were performed similarly to those described previously[69]. In brief, flies were reared at 25 °C on cornmeal medium supplemented with retinal (0.2 mM) that was shielded from light. All experiments were performed on female flies, 3–5 days after eclosion. Brains were dissected in a saline bath (103 mM NaCl, 3 mM KCl, 2 mM CaCl₂, 4 mM MgCl₂, 26 mM NaHCO₃, 1 mM NaH₂PO₄, 8 mM trehalose, 10 mM glucose, 5 mM TES, bubbled with 95% O₂/5% CO₂). After dissection, the brain was positioned anterior side up on a coverslip in a Sylgard dish submerged in 2 ml saline at 20 °C. The sample was imaged with a resonant scanning 2-photon microscope with near-infrared excitation (920 nm, Spectra-Physics,

INSIGHT DS DUAL) and a ×25 objective (Nikon MRD77225 25XW). The microscope was controlled using ScanImage 2017 (Vidrio Technologies). Volumes were acquired with a 230 µm × 230 µm field of view at 512 × 512 px resolution at 2 µm steps over 42 slices, at approximately 1 Hz. The excitation power for Ca²⁺ imaging measurement was 15 mW. On the emission side, the primary dichroic was Di02-R635 (Semrock), the detection arm dichroic was 565DCXR (Chroma), and the emission filters were FF03-525/50 and FF01-625/90 (Semrock). During photostimulation, the light-gated ion channel Chrimson was activated with a 660 nm LED (M660L3 Thorlabs) coupled to a digital micromirror device (Texas Instruments DLPC300 Light Crafter) and combined with the imaging path with a FF757-DiO1 dichroic (Semrock). Photostimulation occurred at 10 Hz over two periods with a duration of 14 s at 0.037 mW mm⁻² intensity interspersed by a 2 s pause. After responses to the photostimulation, the laser power was increased to take two-colour high-resolution images containing fluorescence from both the red and green channels. Using custom Python scripts, regions of interest (ROIs) corresponding to cell compartments were identified in the high-resolution images. These ROIs were then applied to the time-series images to measure intensity changes in response to the photostimulation. Fluorescence in a background ROI, which contained no endogenous fluorescence, was subtracted from the cell compartment ROIs. In the $\Delta F/F$ calculations, baseline fluorescence is the mean fluorescence over a 10 s time period before stimulation started. The $\Delta F$ is the fluorescence minus the baseline. The $\Delta F$ is then divided by the baseline to normalize the signal ($\Delta F/F$). Outlier samples with very low intensities or those of which the intensity randomly fluctuated were excluded from the analysis.

### Electrophysiology

Whole-cell patch-clamp recordings were obtained from freshly isolated brains of 3–5 day old flies. The brain was continuously perfused with oxygenated (95% O₂/5% CO₂) extracellular saline containing 103 mM NaCl, 3 mM KCl, 1.5 mM CaCl₂·2H₂O, 4 mM MgCl₂·6H₂O, 1 mM NaH₂PO₄·H₂O, 26 mM NaHCO₃, 5 mM TES, 10 mM glucose and 10 mM trehalose·2H₂O. Osmolarity was 275 mOsm and the pH was 7.3. Recording electrodes were pulled from thick-walled glass pipette (1.5 mm/0.86 mm) using the P-97 puller (Sutter Instruments) and fire polished using MF 830 (Narishige) to achieve resistances of 10–12 MΩ. Intracellular saline contained 137 mM KAsp, 10 mM HEPES, 1.1 mM EGTA, 0.1 mM CaCl₂·2H₂O, 4 mM MgATP, 0.5 mM NaGTP. Osmolarity was 260–265 mOsm and the pH was adjusted to 7.3 with KOH. Biocytin was added to intracellular solution at 0.5% for post hoc morphological confirmation.

The brain was visualized by an IR-sensitive CCD camera (ThorLabs 1501M) with an 850 nm LED (ThorLabs M850F2). GFP-labelled cell body was visualized with 460 nm LED (Sutter Instruments). Images were acquired using Micro-Manager with automatic contrast adjustment. Recordings were obtained from cell bodies under a 60× water-immersion objective (Olympus).

Current-clamp recordings were sampled at 20 kHz, low-pass filtered at 10 kHz using the Digidata 1550B system, Multiclamp 700B system and Clampex v.11.2 software (Molecular Devices). Recordings were made at a membrane potential of −50 mV to −65 mV, with small (5–30 pA) hyperpolarizing current injections as needed, and not corrected for liquid junction potentials.

Chrimson was activated by 630 nm LED at 0.4 mW cm⁻². The stimulation duration was set at minimal value that is sufficient to induce reliable responses from target neurons. After the electrophysiology recording, the whole brain was fixed in 4% paraformaldehyde in 0.1 M PBS until further staining. After rinsing in PBS, the brain was incubated in Streptavidin Alexa Fluor 647 (1:200) in PBS-T overnight at room temperature. The preparations were then rinsed, dehydrated and mounted with DPX. The confocal images were captured on the LSM 980 microscope (Zeiss), with 639 nm excitation wavelength.

The electrophysiological recordings were analysed using pClamp (Clampfit v.11.3). The instantaneous action potential frequency was

calculated for about 1 min in each cell. The action potential amplitude was averaged from 20–30 individual events in each cell, and measured as the difference between the threshold and peak.

## Immunohistochemistry and imaging

All experiments were performed as described previously[46,70–74]. Additional details of the imaging pipeline used are available online (https://data.janelia.org/pipeline).

## Connectomics analyses

Our analyses are based on the hemibrain dataset[25] (v.1.2.1) as queried using the neuPrint interface (https://neuprint.janelia.org/) unless otherwise noted. The unique identifier (bodyID number in the hemibrain v.1.2.1 database) for neurons is shown in the figures, and a complete list of synaptic connections used to construct our circuit diagrams can be found in neuPrint. LC10s in the hemibrain dataset were assigned to candidate types based on connectivity and morphology differences. As the hemibrain did not include the entire lobula, we also performed analyses in the recently completed and fully annotated male optic lobe connectome[53] (neuprint release: optic-lobe: v.1.0) and the Flywire[54] (v.783) analysis of the FAFB datasets[75].

Synaptic connections organized by neuron pair and neuropil for Fly-Wire or optic lobe were obtained from Zenodo[76] and neuPrint (using neuprint-python fetch_adjacencies with the default settings), respectively. The connection tables obtained in this way do not include synapses with small, untyped EM bodies as, in most cases, unidentified fragments of larger reconstructed cells are included in the tables and we did not include such synapses when calculating relative connection strengths. Including these additional synapses in calculations of relative connections strength (as done for example in the neuPrint web interface) results in lower numbers that most likely underestimate the relative contribution of one cell type to the inputs or outputs of another type. Although we did observe highly similar relative connection strengths for many cell type pairs, we also note that there were methodological differences between the FlyWire/FAFB and optic lobe datasets that may result in significant non-biological differences between synapse counts.

Neurotransmitters for TuTuA_1 and IB112 were determined by EASI-FISH[77] using lines SS77547 for TuTuA_1 and SS81529 for IB112. Neurotransmitters for aIPg[46] and LC10s[78] were previously reported. Other reported neurotransmitters are based on computational predictions[53,79].

Cell type annotations for FlyWire neurons[59,60] were as downloaded from Codex on 24 April 2024 from https://codex.flywire.ai/api/download?data_product=visual_neuron_types&data_version=783. Matches of cell type names between the optic lobe and FlyWire datasets were as published[53]. FlyWire data to evaluate connections to descending interneurons described in Extended Data Fig. 10 were accessed through Codex (https://codex.flywire.ai) and evaluated using Flywire.

## Statistics

No statistical methods were used to predetermine sample size. Sample size was based on previous literature in the field and is provided in the Supplementary Information (Supplementary Tables 6, 8, 12, 14, 16, 19, 23, 25, 31 and 35). Experimenters were not blinded in most conditions as all data acquisition and analysis were automated. Experiments were not randomized as most controls were performed within animals. Biological replicates completed at separate times using different parental crosses were performed for each of the behavioural experiments. Behavioural data are representative of at least two independent biological repeats. For figures in which the behavioural data over the course of a trial are shown, a yellow or red bar indicates the stimulus period, the mean is represented as a solid line and shaded error bars represent the variation between experiments.

For each experiment, the experimental and control flies were collected, treated and tested at the same time. A Wilcoxon matched-pairs signed-rank test (two-tailed) was used for statistical analysis of optogenetic experiments when examining effects within the same group. The results of two-way analysis of variance followed by Tukey's test (three or more groups) or uncorrected Fisher's least significant difference test (two groups) for multiple comparisons for each optogenetic experiment are also reported in the Supplementary Information. For analysis among two groups, Mann–Whitney $U$-tests (two-tailed) were used, while Kruskal–Wallis tests with Dunn's multiple-comparisons post hoc analysis were used to compare across multiple groups. All statistical analysis was performed using Prism (GraphPad, v.10). $P$ values are indicated by asterisks. The exact $P$ values for each figure are provided in the Supplementary Information.

For bar plots, all datapoints are shown to indicate the range and the top edge of the bar represents the mean. The box plots show the median and interquartile range. The lower and upper whiskers represent 1.5 × interquartile range of the lower and upper quartiles, respectively; the boxes indicate lower quartile, median and upper quartile, from bottom to top. When all points are shown, the whiskers represent the range and the boxes indicate the lower quartile, median and upper quartile, from bottom to top. In the violin plots, the lower and upper quartiles are indicated by dotted light grey lines, and the median is indicated by a solid light grey line. Shaded error bars on graphs represent the mean ± s.e.m.

## Reporting summary

Further information on research design is available in the Nature Portfolio Reporting Summary linked to this article.

## Data availability

All data underlying this study are included in the Article and are publicly available at Figshare[80] (https://doi.org/10.25378/janelia.26847772). Any additional information is available on request from the corresponding author. Source data are provided with this paper.

## Code availability

Visual features and behavioural analysis code are available at Figshare[81] (https://doi.org/10.25378/janelia.26849083) and GitHub (https://github.com/ceschretter/SocialState2024_Code). Any additional information is available on request from the corresponding author.

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

**Acknowledgements We** thank L. Abbott, B. Mensh and the Janelia Community for their suggestions during the course of this work and their comments on the manuscript; the members of the Fly Light team for generating the images of GAL4 expression patterns; K. Hibbard and H. Dionne for assisting with generation of genetic reagents for female behavioural experiments, and Project Technical Resources for performing EASI-FISH experiments. This work was supported by Howard Hughes Medical Institute, Simons Foundation Collaboration on the Global Brain and by an NIH NINDS grant (5R35NS111611) to V.R.

**Author contributions** C.E.S. and G.M.R. conceived and designed the study. G.M.R., M.D. and A.N. performed connectome analyses. C.E.S. performed strain construction as well as female behavioural experiments and analysis. G.M.R. generated cell-type-specific genetic driver lines with assistance from C.E.S. and the Janelia Fly Light Project Team. The behavioural analysis pipeline was designed and developed by K.B., A.A.R., A.L.T. and C.E.S. N.K. performed functional imaging experiments shown in Fig. 3a and Extended Data Fig. 2a–d. C.E.S. and A.O. performed computational analysis of visual parameters during female aggression and male courtship in Fig. 3d and Extended Data Fig. 2e–f. Electrophysiology recordings and analysis were performed by M.S. D.B. performed in vitro functional imaging experiments shown in Extended Data Figs. 5b and 7a–e. T.H.S. and V.R. designed, performed and analysed the functional imaging and male courtship experiments in Fig. 6c. M.D., C.E.S., T.H.S. and A.O. prepared graphics for figures. The original draft of the manuscript was written by C.E.S. and G.M.R. with input from all of the authors.

**Competing interests** The authors declare no competing interests.

**Additional information**
**Correspondence and requests for materials** should be addressed to Gerald M. Rubin.

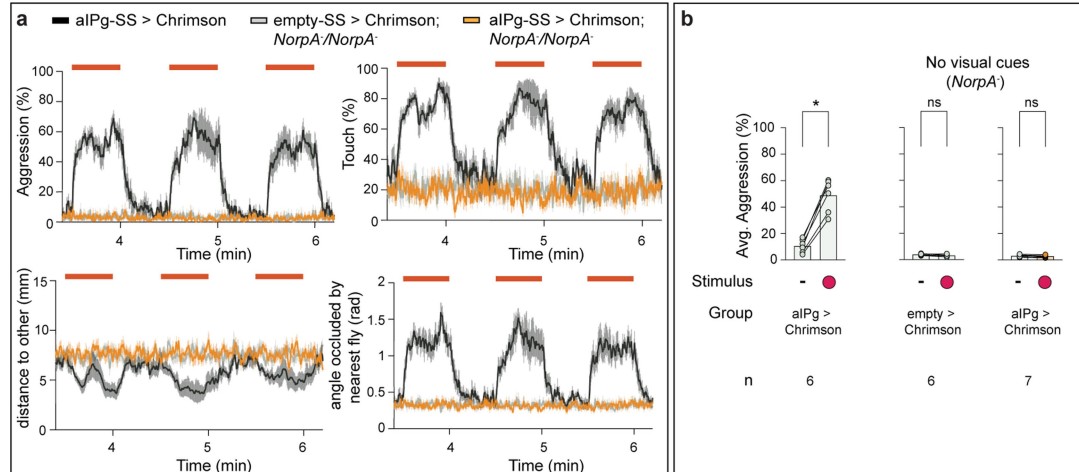

**Extended Data Fig. 1 | Pathways carrying visual information are important for female aggressive behaviours and related behavioural features.** (**a**) Percentage of flies engaging in aggressive behaviours, touching, and changes in parameters related to distance to another fly and the maximum angle of the field of view occluded by the closest fly (angle occluded by nearest fly) are plotted over the course of a trial during which a three 30 s 3 mW/cm$^2$ continuous red-light stimulus (red bars) were delivered. Low intensity stimuli (1 mW/cm$^2$; not shown) produced lower levels of aggression in the aIPg-SS > Chrimson group and no significant changes in the no visual cues (NorpA$^{-/-}$) groups. The mean is represented as a solid line and shaded bars represent standard error between experiments. (**b**) Average time spent performing aggressive behaviours before and during stimulus periods. All data points are shown to indicating the range and top edge of bar represents the mean. Each dot represents one experiment containing approximately seven flies. Data supporting the plots shown in panels a–b were as follows: aIPg-SS > Chrimson, n = 6 experiments; NorpA$^{-/-}$ EmptySS > Chrimson, n = 6 experiments; NorpA$^{-/-}$ aIPg-SS > Chrimson, n = 7 experiments. Data are representative of two biological replicates, which included separate parental crosses and were collected on different days. A non-parametric Wilcoxon Matched-pairs Signed Rank test was used for statistical analysis. Asterisk indicates significance from 0: *p < 0.05.

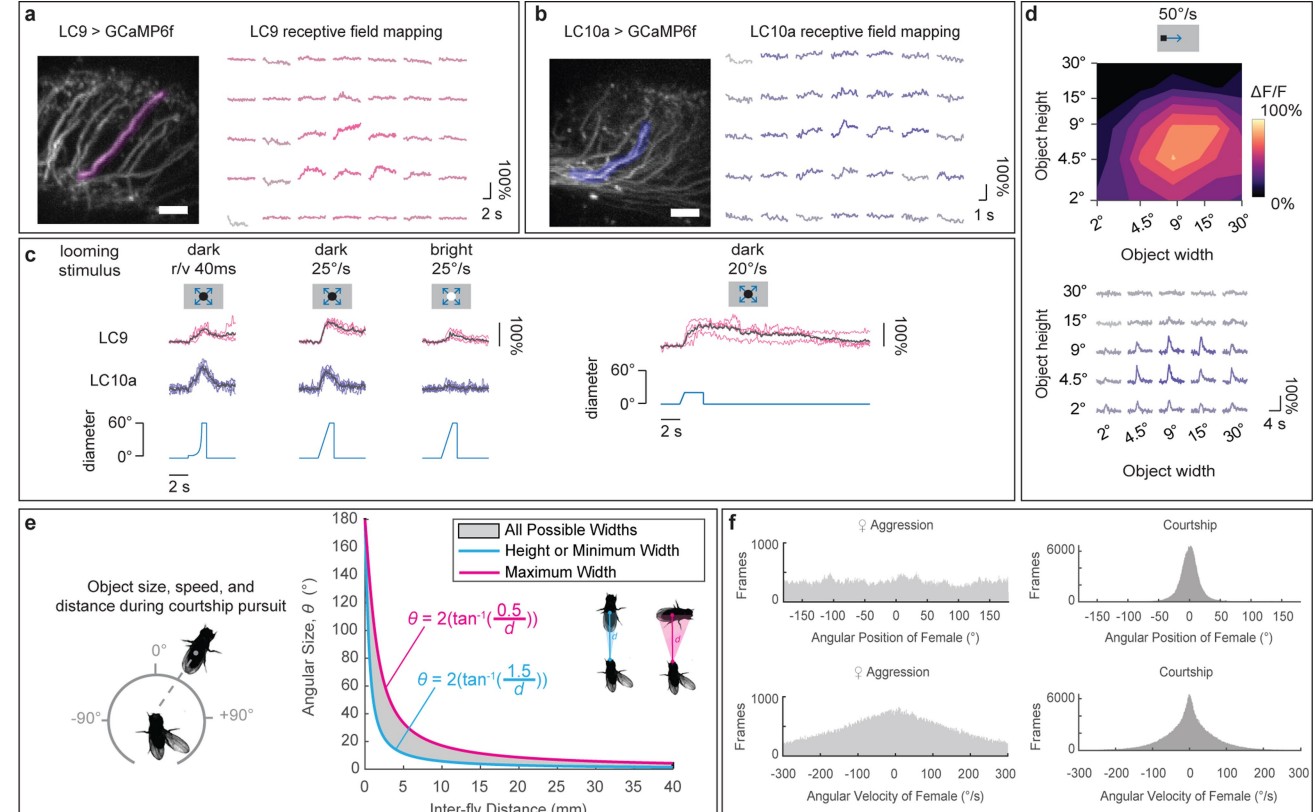

**Extended Data Fig. 2 | LC visual feature detection.** Panels a–d show the receptive fields for LC9 and LC10a. Panels e – f show the visual experience during male courtship and female aggression. **(a–b)** Single receptive-field mapping for individual LC axons in representative flies. LC9 and LC10a axonal regions of interest are coloured in magenta and blue, respectively and overlaid on averaged calcium image (left; scale bar: 10 µm). Individual calcium responses, arranged as in Fig. 3a, are shown on right. **(c)** Single-cell (colour) and population average (black) calcium traces for neurons responding to looming stimuli centred on the receptive field, same as performed in[24]. LC9: n = 4 female flies, n = 4 neurons. LC10a: n = 5 male flies, n = 7 neurons (25°/s constant edge speed looming was only recorded for 2 neurons from 1 fly). **(d)** Size tuning, as measured and plotted in Fig. 3a, for objects of varying sizes moving at a slower speed of 50°/s. **(e–f)** (e) Histogram shows conspecific angular position in the visual field as experienced by the male during courtship pursuit. All possible angular heights and widths for a female with a minor axis of 1 mm and major axis of 3 mm are plotted. (f) Histograms of angular position and velocity during aIPg-mediated female aggression (left) and naturalistic male courtship pursuit (right).

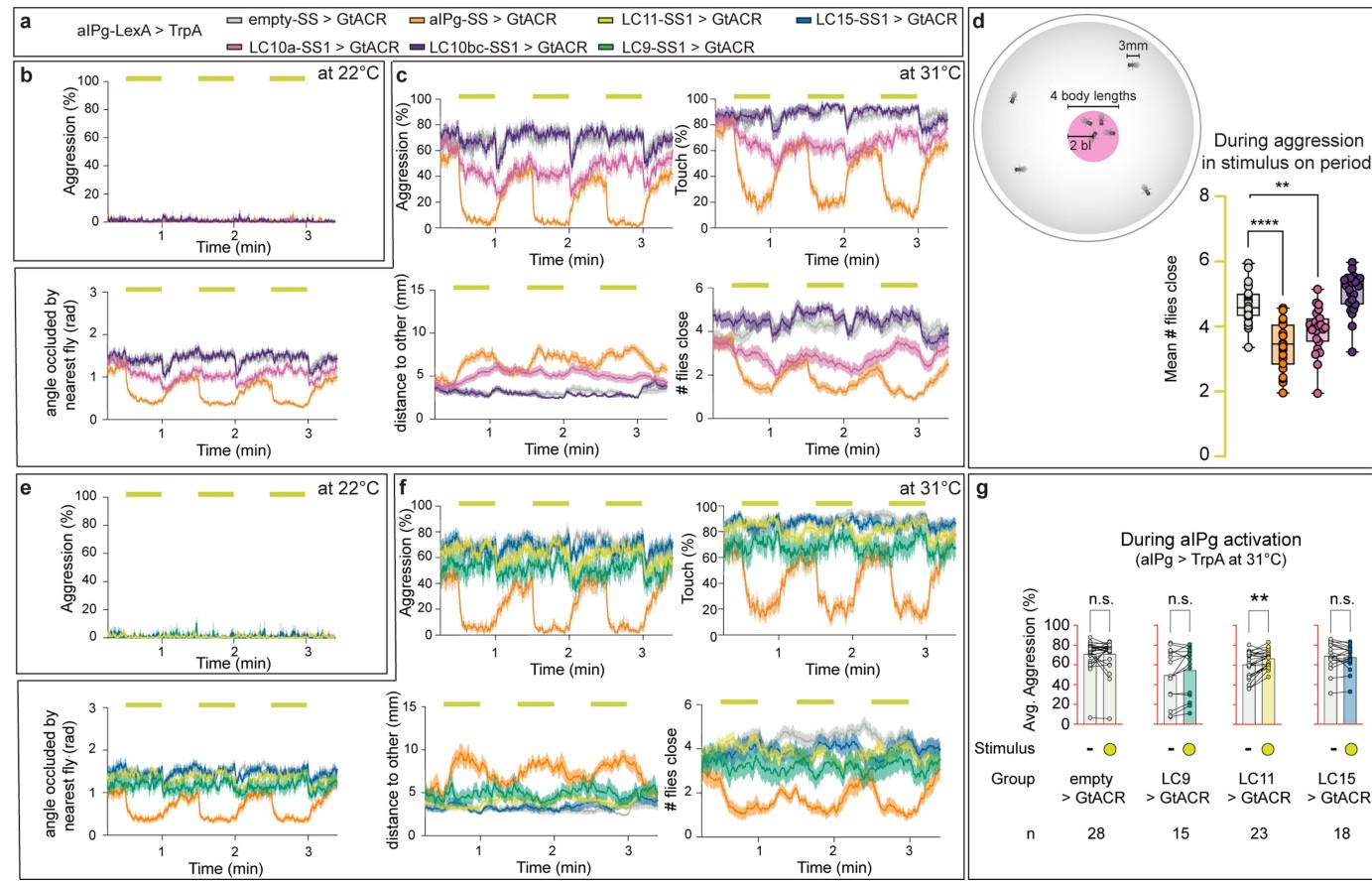

**Extended Data Fig. 3 | LC involvement in female aggressive behaviours.**
**(a)** Legend for panels b–g. **(b–c, e–f)** Percentage of flies engaging in behaviours (aggression, touch) or behavioural features (distance to other, angle occluded by nearest fly, number of flies close) over the course of a trial during which three 30 s continuous light stimuli (yellow bars) were delivered. To control for additional cell types in the LexA line used for aIPg[82], we simultaneously inhibited aIPg during thermogenetic activation through using an aIPg-specific split-GAL4 line and the green light gated anion channel, GtACR. The dramatic reduction in female aggressive behaviour during optogenetic inhibition confirmed that aIPg was primarily responsible for the aggression observed when stimulating the LexA > TrpA line. **(d)** The mean number of flies close (pink region in the diagram shown in inset image, 1 body length = ~3 mm) was calculated during the aggression bouts performed in the stimulus periods shown in c. **(g)** Average time spent performing aggressive behaviours before and during stimulus periods in which a 30 s continuous green (9 mW/cm²) light stimulus was delivered (stimulus on, green dot). All data points are shown to indicate the range and the top edge of bar represents the mean. Data were pooled from eight biological replicates, which included separate parental crosses and were collected on different days. Data supporting the plots shown

in panels b – e were as follows: b: aIPg-LexA > TrpA emptySS > GtACR, n = 16 experiments; aIPg-LexA > TrpA aIPg-SS > GtACR, n = 11 experiments; aIPg-LexA > TrpA LC10a-SS > GtACR, n = 16 experiments; aIPg-LexA > TrpA LC10bc-SS > GtACR, n = 10. c – d: aIPg-LexA > TrpA emptySS > GtACR, n = 26 experiments; aIPg-LexA > TrpA aIPg-SS > GtACR, n = 26 experiments; aIPg-LexA > TrpA LC10a-SS > GtACR, n = 23 experiments; aIPg-LexA > TrpA LC10bc-SS > GtACR, n = 23. e: aIPg-LexA > TrpA emptySS > GtACR, n = 19 experiments; aIPg-LexA > TrpA aIPg-SS > GtACR, n = 15 experiments; aIPg-LexA > TrpA LC9-SS > GtACR, n = 6 experiments; aIPg-LexA > TrpA LC11-SS > GtACR, n = 12; aIPg-LexA > TrpA LC15-SS > GtACR, n = 5. experiments. Experiments were performed at a temperature that activates TrpA (31 °C) in c – d, f – g for aIPg > TrpA stimulation; and non-activating temperature controls (22 °C) are shown in b and e. The mean is represented as a solid line and shaded bars represent standard error between experiments. The timeseries shows the percentage of flies performing aggression displayed as the mean of 0.35 s (60-frame) bins. Box-and-whisker plots show median and IQR; whiskers show range. A Kruskal-Wallis and Dunn's post hoc test (d) or non-parametric Wilcoxon Matched-pairs Signed Rank test (g) were used for statistical analysis. Asterisk indicates significance from 0: *p < 0.05, **p < 0.01, ****p < 0.0001.

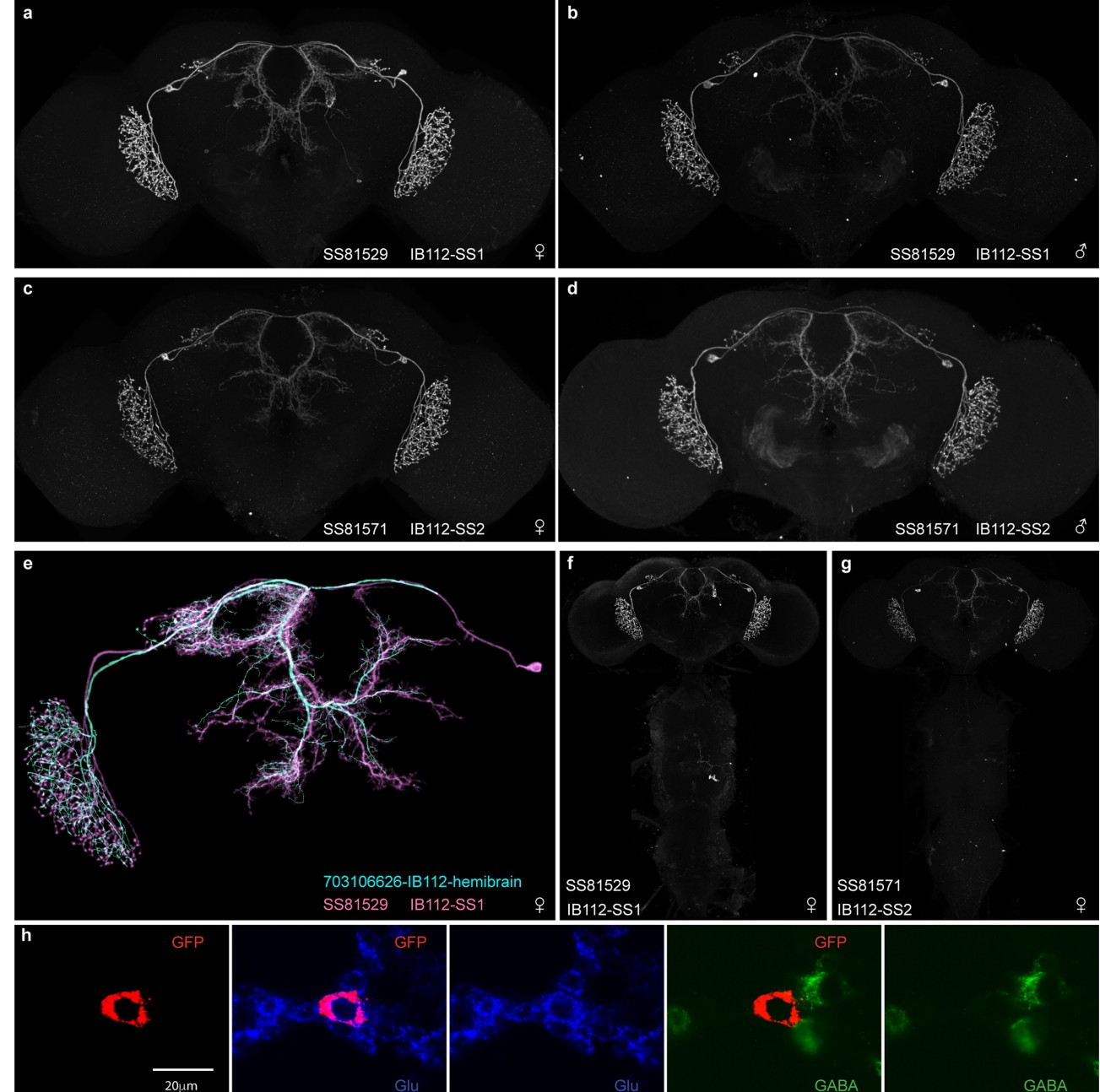

**Extended Data Fig. 4 | Anatomy of GAL4 driver lines for IB112.**
(**a, b**) Expression patterns in a female and male brain, respectively, of GAL4 line SS81529 (IB112-SS1). (**c, d**) Expression patterns in a female and male brain, respectively, of GAL4 line SS81571 (IB112-SS2). (**e**) IB112 body ID 703106626 skeleton from hemibrain v1.2.1 shown together with a neuron from SS81529 obtained by stochastic labelling[70] and then segmented using VVD viewer (see Supplementary Table 1). (**f, g**) Images of the expression patterns in the brain and VNC of GAL4 driver lines SS81529 and SS81571, as indicated. (**h**) Images of fluorescent in situ hybridization assays to determine the neurotransmitter used by IB112. Probes used in each panel are indicated. GFP shows the IB112 cell body and Glu and GABA represent probes for vGlut and GAD, respectively (see Methods for details). Scale bar is shown in the left panel.

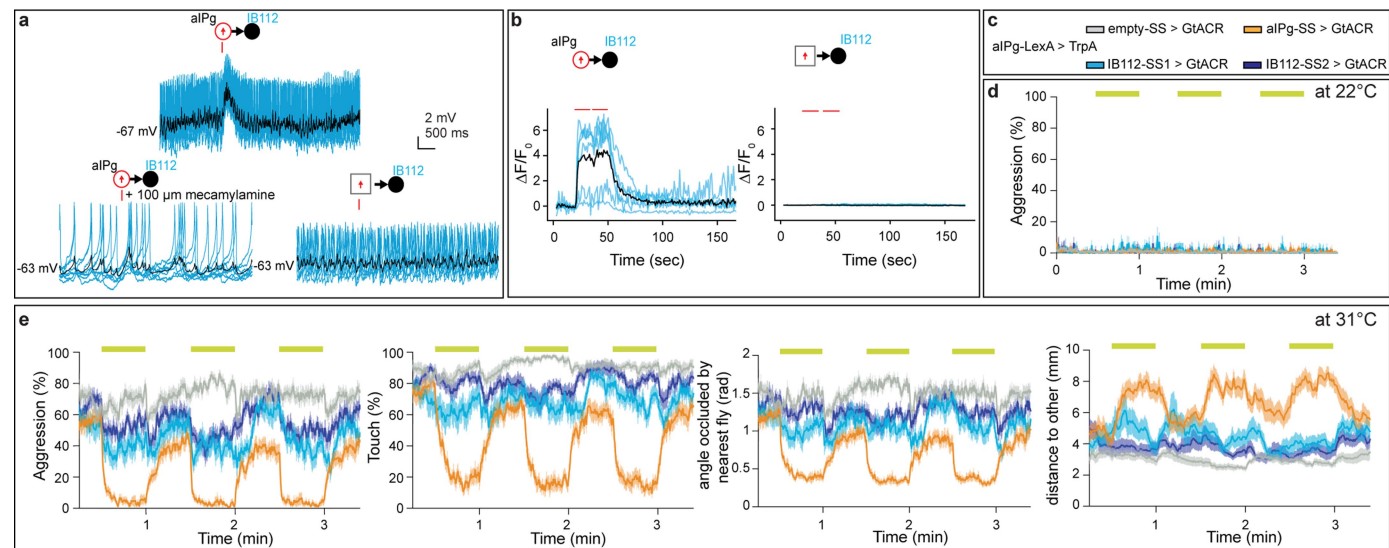

**Extended Data Fig. 5 | IB112 shapes aIPg-mediated female aggressive behaviours.** (**a**) Excitatory responses recorded by patch clamp electrophysiology in female brain explants from IB112 (n = 6 cells) before, during, and following a 15 ms activation of aIPg. Individual trials in blue (n = 8 trials from one cell), mean shown in black. No evoked response was recorded when stimulating Chrimson in the presence of mecamylamine or in the absence of a LexA driver (n = 5, bottom right). (**b**) Changes in fluorescence intensity as measured by GCaMP6f in the cell body of IB112 to two 14 s optogenetic stimuli (2 s interval) at 10 Hz (n = 6). No changes in fluorescence intensity were recorded when stimulating with Chrimson alone (n = 3). Individual trials are shown in blue and the mean is in black. (**c–e**) Percentage of flies engaging in aggression, touch, or changes in related parameters, including the maximum angle of the field of view occluded by the closest fly (angle occluded by nearest fly) or distance to another fly. Legend for figures d–e is shown in c. Percentages are plotted over the course of a trial during which three 30 s 9 mW/cm² continuous light stimuli (yellow bars) were delivered. The mean is represented as a solid line and shaded bars represent standard error between experiments. The timeseries shows the percentage of flies performing aggression displayed as the mean of 0.35 s (60-frame) bins. Data supporting the plots shown in panels d–e were as follows: d: aIPg-LexA > TrpA emptySS > GtACR, n = 17 experiments; aIPg-LexA > TrpA aIPg-SS > GtACR, n = 10 experiments; aIPg-LexA > TrpA IB112-SS1 > GtACR, n = 4 experiments; aIPg-LexA > TrpA IB112-SS2 > GtACR, n = 5 experiments. e: aIPg-LexA > TrpA emptySS > GtACR, n = 22 experiments; aIPg-LexA > TrpA aIPg-SS > GtACR, n = 22 experiments; aIPg-LexA > TrpA IB112-SS1 > GtACR, n = 17 experiments; aIPg-LexA > TrpA IB112-SS2 > GtACR, n = 20 experiments. Experiments performed at a temperature that activates TrpA (31 °C) e for aIPg > TrpA stimulation, and non-activating temperature controls (22 °C), are shown in d. Data were pooled from four biological replicates, which included separate parental crosses and were collected on different days.

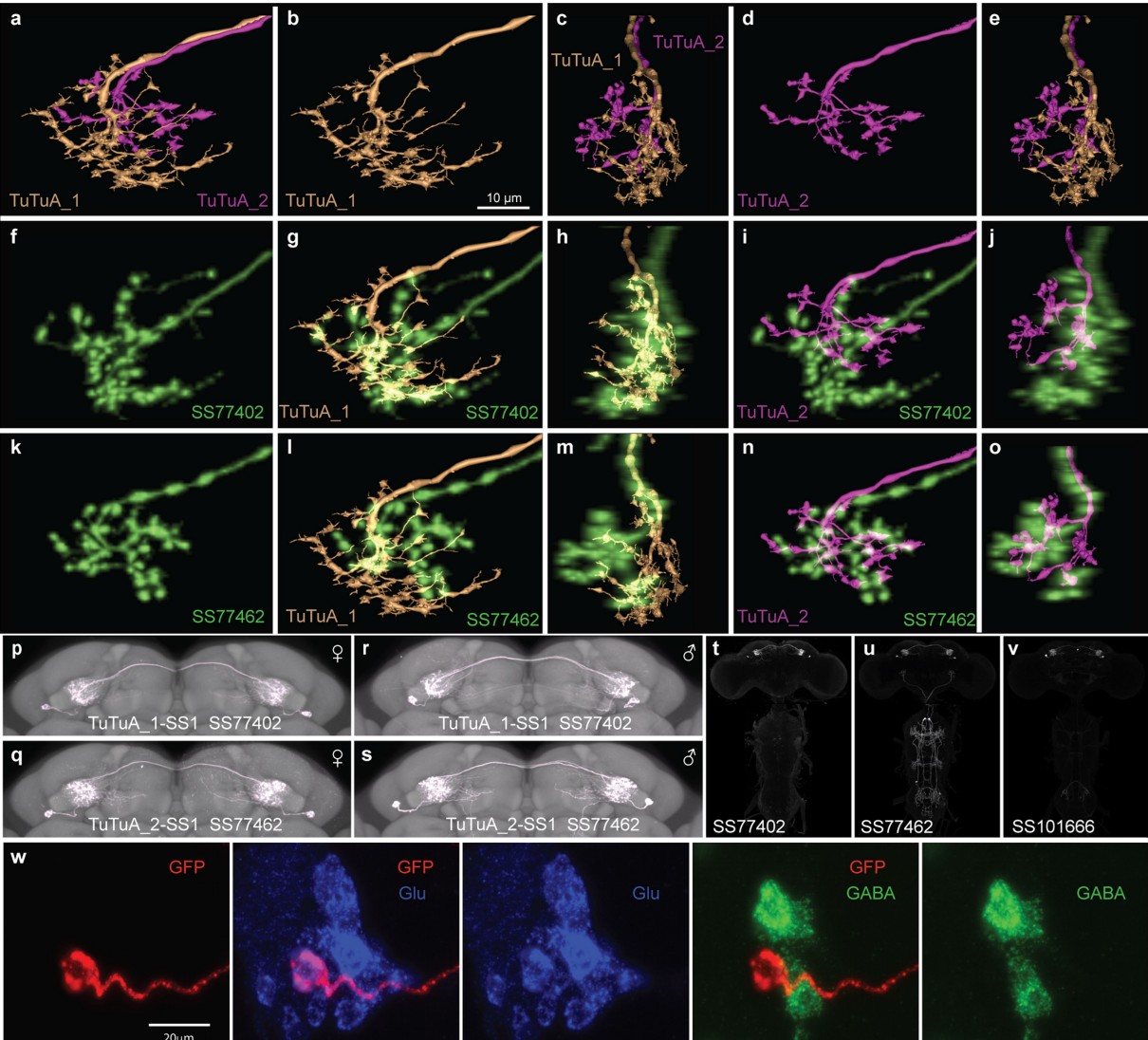

**Extended Data Fig. 6 | Anatomy of TuTuA cell types.** (**a–e**) Neuronal skeletons of the termini of the contralateral axons of TuTuAs from the hemibrain v1.2.1 connectome. (a) TuTuA_1 and TuTuA_2 are shown together (body IDs 676836779 and 5813013691, respectively). (b) TuTuA_1 (body ID 676836779) shown alone. (c) Same as panel a, but rotated 90 degrees along the medial-lateral axis. (d) TuTuA_2 (body ID 5813013691) shown alone. (e) Same as panel c, repeated to facilitate comparison. These anatomical differences were used to determine the correspondence between GAL4 driver lines and TuTuA subtypes. (**f**) Terminus of a contralateral axon of a neuron from GAL4 driver line SS77402 obtained by stochastic labelling[70] and then segmented using VVD viewer (see Supplementary Table 1). (**g**) The comparison between the GAL4 driver line in f to TuTuA_1 skeleton shown in b. (**h**) Same as panel g, but rotated 90 degrees along the medial-lateral axis. (**i**) The comparison between the GAL4 driver line in f to TuTuA_2 skeleton shown in d. (**j**) Same as panel i but rotated 90 degrees along the medial-lateral axis. (**k**) Terminus of a contralateral axon of a neuron from GAL4 driver line SS77462 obtained by stochastic labelling[70] and

then segmented using VVD viewer. (**l**) The comparison between the GAL4 driver line in k to TuTuA_1 skeleton shown in b. (**m**) Same as panel g, but rotated 90 degrees along the medial-lateral axis. (**n**) The comparison between the GAL4 driver line in k to TuTuA_2 skeleton shown in d. (**o**) Same as panel n, but rotated 90 degrees along the medial-lateral axis. (**p, r**) Images of GAL4 driver line SS77402 in females and males, respectively, shown with the standard neuropil reference, JFRC2018U[83]. Note the presence of a single TuTuA cell body in each brain hemisphere. (**q, s**) Images of GAL4 driver line SS77462 in females and males, respectively. Note the presence of a single TuTuA cell body in each brain hemisphere. (**t–v**) Images of the expression patterns in the brain and VNC of GAL4 driver lines SS77402, SS77462 and SS10166, as indicated. (**w**) Images of fluorescent in situ hybridization assays to determine the neurotransmitter used by TuTuA_1. Probes used in each panel are indicated. GFP shows the TuTuA_1 cell and Glu and GABA represent probes for vGlut and GAD, respectively (see Methods for details). Scale bar is shown in the left panel.

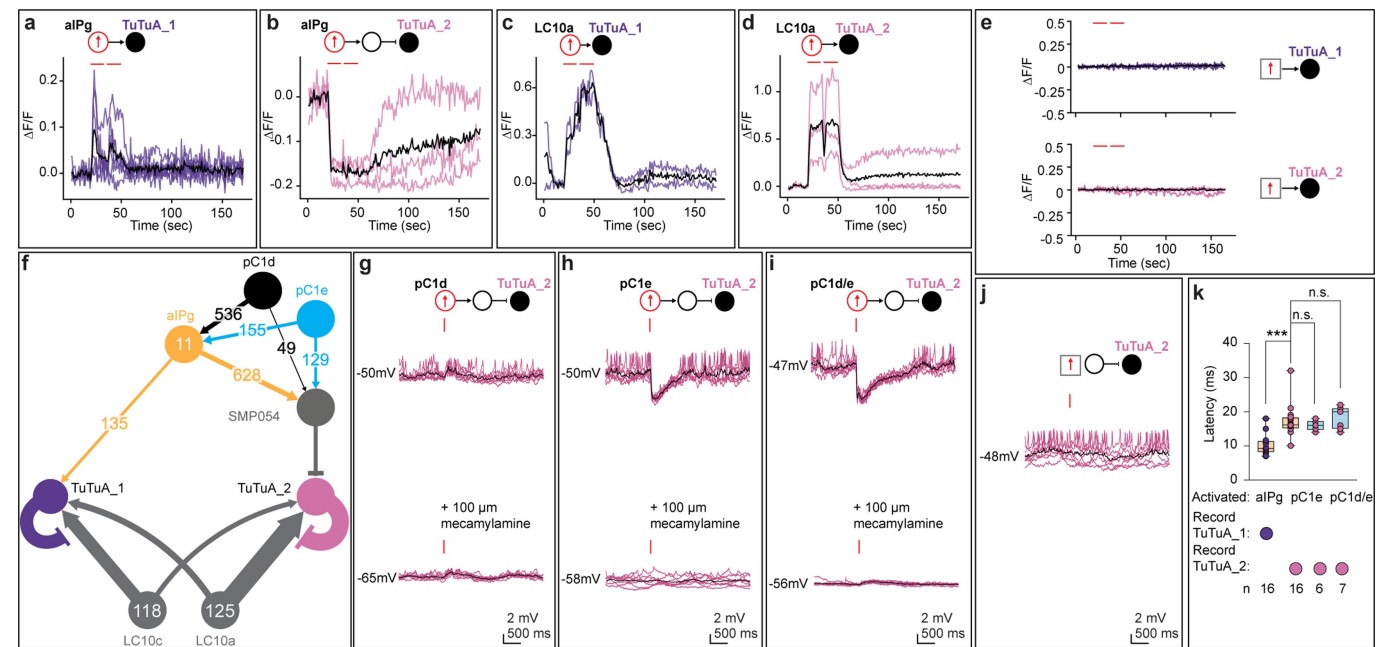

**Extended Data Fig. 7 | Responses of TuTuA subtypes to the activation of female aggression or LC10a cell types. (a–e)** Changes in fluorescence intensity as measured by GCaMP6f in the cell body of TuTuA_1 (a, c) or TuTuA_2 (b, d) in response to two 14 s optogenetic stimuli (2 s interval) at *10 Hz* (n = 3 – 4). (e) No changes in fluorescence intensity were recorded when stimulating with Chrimson alone (n = 4). Individual trials for a – e are shown in purple (TuTuA_1) or pink (TuTuA_2), mean is in black. **(f)** Connectivity diagram from Fig. 5a with the connections from pC1d and pC1e. Synapse numbers are indicated on the arrows, which are also scaled according to synapse counts. Arrows indicate putative excitatory connections (cholinergic) and bar endings indicate putative inhibitory connections (SMP054, GABAergic; TuTuA_1 and TuTuA_2, glutamatergic). **(g–j)** Electrophysiology recordings with the cell types activated with Chrimson are circled in red, and those recorded are in black. Individual trials are in pink (n = 8 trials from 1 cell), mean is in black. (g) Small excitation or negligible response in TuTuA_2 (n = 5 cells) to 15 ms pC1d activation, which was abolished by mecamylamine. (h) Large inhibitory response in TuTuA_2 to 15 ms pC1e activation, which was abolished by mecamylamine (n = 6 cells). (i) Large inhibitory response in TuTuA_2 to 15 ms pC1d/e activation, which was abolished by mecamylamine (n = 8 cells). (j) No evoked response was recorded when stimulating Chrimson in the absence of a LexA driver (n = 6). **(k)** Latency after stimulus onset (ms). Box-and-whisker plots show median and IQR; whiskers show range. A Kruskal-Wallis and Dunn's post hoc test was used for statistical analysis. Asterisk indicates significance from 0: ***p < 0.001.

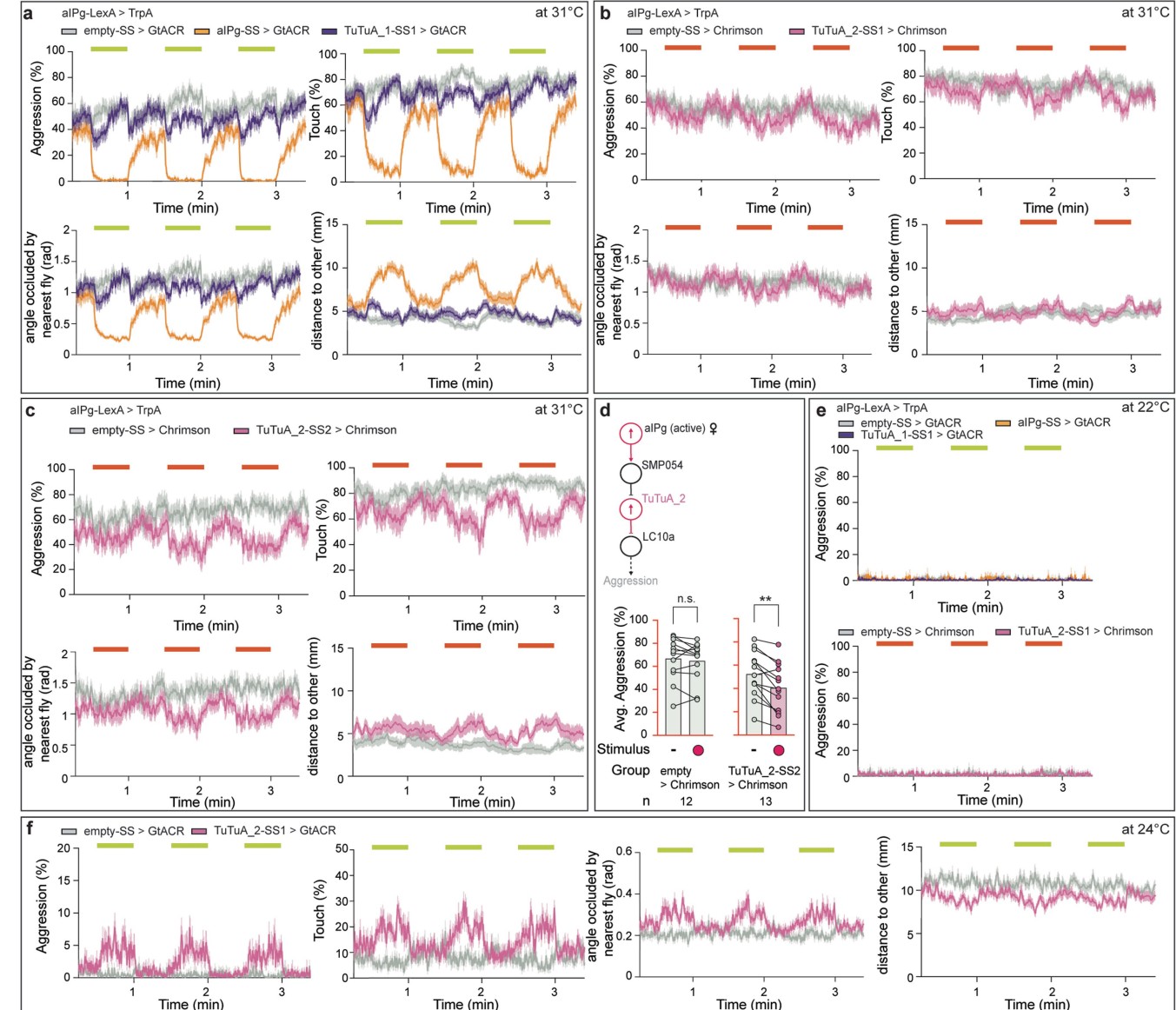

**Extended Data Fig. 8 | The TuTuA switch shapes female aggressive behaviours.** (**a–c, e–f**) Percentage of flies engaging in behaviours (aggression, touch) or behavioural features (distance to other, angle occluded by nearest fly) over the course of a trial during which three 30 s continuous light stimuli (yellow or red bars) were delivered. Experiments were performed at a temperature that activates TrpA (31 °C, a–c) for aIPg > TrpA stimulation; and non-activating temperature controls (22 °C) are shown in e. (**d**) Percentage of flies engaging in aggression over the course of a trial during which three 30 s continuous 3 mW/cm² red light (red bars) were delivered. Data were pooled from three biological replicates, which included separate parental crosses and were collected on different days. Data supporting the plots shown in panels a–f were as follows: a: aIPg-LexA > TrpA emptySS > GtACR, n = 23 experiments; aIPg-LexA > TrpA aIPg-SS > GtACR, n = 22 experiments; aIPg-LexA > TrpA TuTuA_1-SS > GtACR, n = 24 experiments. b: aIPg-LexA > TrpA emptySS > Chrimson, n = 19 experiments; aIPg-LexA > TrpA TuTuA_2-SS1 > Chrimson,

n = 20 experiments. c, d: aIPg-LexA > TrpA emptySS > Chrimson, n = 12 experiments; aIPg-LexA > TrpA TuTuA_2-SS2 > Chrimson, n = 13 experiments. e (top panel): aIPg-LexA > TrpA emptySS > GtACR, n = 14 experiments; aIPg-LexA > TrpA aIPg-SS > GtACR, n = 5 experiments; aIPg-LexA > TrpA TuTuA_1-SS, n = 14 experiments. e (bottom panel): aIPg-LexA > TrpA emptySS > Chrimson, n = 8 experiments; aIPg-LexA > TrpA TuTuA_2-SS1 > Chrimson, n = 11 experiments. f: emptySS > GtACR, n = 20 experiments; TuTuA_2-SS1 > GtACR, n = 20 experiments. The mean for a–c and d–f is represented as a solid line and shaded bars represent standard error between experiments. The timeseries shows the percentage of flies performing aggression displayed as the mean of 0.35 s (60-frame) bins. Averages were calculated over all flies in an experiment, with each dot representing one experiment containing approximately seven flies. All data points are shown to indicating the range and top edge of bar represents the mean. A non-parametric Wilcoxon Matched-pairs Signed Rank test (d) was used for statistical analysis. Asterisk indicates significance from 0: **p < 0.01.

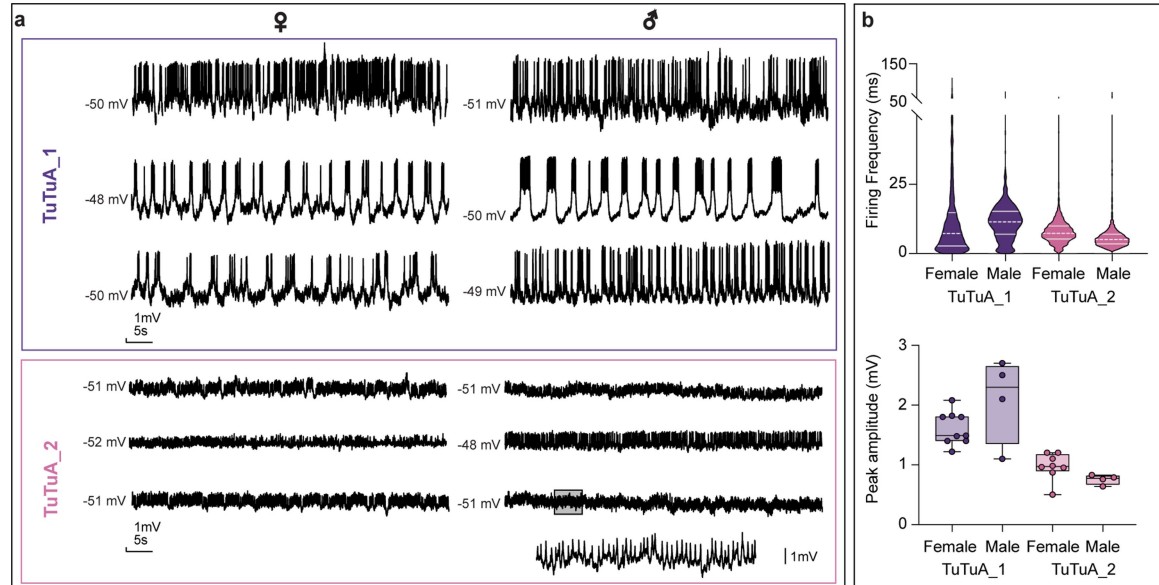

**Extended Data Fig. 9 | Recordings from TuTuA_1 and TuTuA_2 in males and females.** (**a**) Each trace is one-minute recording from one cell. TuTuA_1 displayed the larger action potential amplitude compared to TuTuA_2. Inset recording from a TuTuA_2 neuron is from the highlighted region of the last male recording. (**b**) Analysis of the firing frequency and peak amplitude of TuTuA_1 and TuTuA_2 recordings in males and females. The instantaneous action potential frequency was calculated for about one minute in each cell (TuTuA_1: Female, n = 1536, Male, n = 1965; TuTuA_2: Female, n = 1198, Male, n = 1185). The action potential amplitude was averaged from 20–30 individual events in each cell (each dot represents 1 cell) and measured as the difference between the threshold and peak (TuTuA_1: Female, n = 9, Male, n = 4; TuTuA_2: Female, n = 8, Male, n = 4). The firing frequency was more variable in the TuTuA_1 recordings than in the TuTuA_2 recordings in both males and females. Additionally, the amplitude from TuTuA_1 was larger compared to TuTuA_2 in both males and females. However, the action potential is dramatically slower in male TuTuA_2 neurons. Box-and-whisker and violin plots show median and IQR; whiskers or ends of the violin show range.

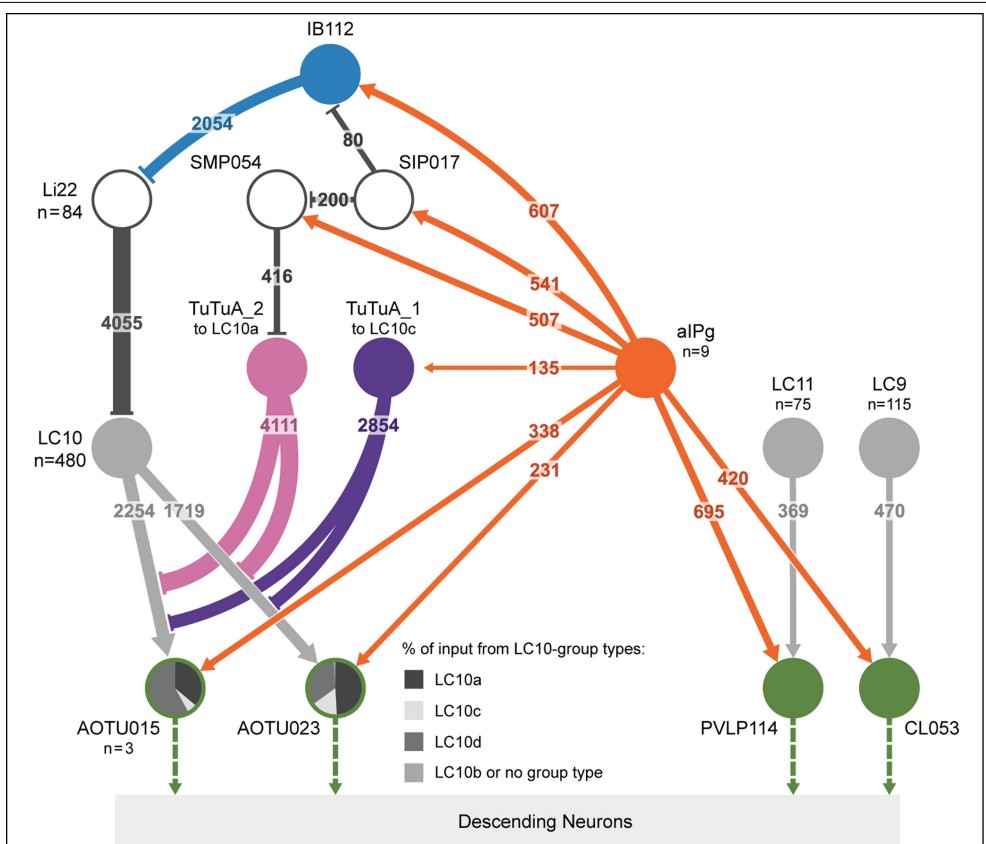

**Extended Data Fig. 10 | Circuit diagram of aIPg modulation of visual processing.** A detailed circuit map of the mechanisms detailed in Fig. 1b. This diagram shows additional downstream targets of aIPg including those involved in regulating information flow from LC9 and LC11. The diagram also illustrates that CL053, PVLP114, AOTU015 and AOTU023 are each upstream of descending interneurons (DNs) that traverse the neck into the ventral nerve cord where they likely regulate motor action. Each of these neurons connect to largely non-overlapping sets of DNs[51,84,85], implying that these parallel pathways control different motor actions. Cell numbers are listed under cell type name and the number of synapses between cell types are listed on the arrows.

Numbers within arrows indicate synapses numbers. The top six downstream targets of aIPg are represented in this diagram: (1) PVLP114; (2) IB112; (3) SMP054; (4) SIP017; (5) CL053; and (6) AOTU015. In the male OL[53], Li22 devotes about 40% of its output synapses to LC10-group cell types and provides input to all LC10-group cell types. On average, the number of Li22 inputs to individual cells in each of the LC10-group cell types are as follows: LC10a, 23; LC10b, 12; LC10c, 27; LC10d, 32; LC10e, 4; and LoVP76/LC10f, 0 in males and LC10a, 8; LC10b, 6; LC10c, 14; LC10d, 15; LC10e, 1.4 and LC10f 1.5 in females. Synapse counts shown are from hemibrain data except IB112 to Li22 and Li22 to LC10s, which were from FlyWire.

# Reporting Summary

## Statistics

For all statistical analyses, confirm that the following items are present in the figure legend, table legend, main text, or Methods section.

| n/a | Confirmed | |
|---|---|---|
| ☐ | ☒ | The exact sample size (*n*) for each experimental group/condition, given as a discrete number and unit of measurement |
| ☐ | ☒ | A statement on whether measurements were taken from distinct samples or whether the same sample was measured repeatedly |
| ☐ | ☒ | The statistical test(s) used AND whether they are one- or two-sided *Only common tests should be described solely by name; describe more complex techniques in the Methods section.* |
| ☐ | ☒ | A description of all covariates tested |
| ☐ | ☒ | A description of any assumptions or corrections, such as tests of normality and adjustment for multiple comparisons |
| ☐ | ☒ | A full description of the statistical parameters including central tendency (e.g. means) or other basic estimates (e.g. regression coefficient) AND variation (e.g. standard deviation) or associated estimates of uncertainty (e.g. confidence intervals) |
| ☐ | ☒ | For null hypothesis testing, the test statistic (e.g. *F*, *t*, *r*) with confidence intervals, effect sizes, degrees of freedom and *P* value noted *Give P values as exact values whenever suitable.* |
| ☒ | ☐ | For Bayesian analysis, information on the choice of priors and Markov chain Monte Carlo settings |
| ☒ | ☐ | For hierarchical and complex designs, identification of the appropriate level for tests and full reporting of outcomes |
| ☒ | ☐ | Estimates of effect sizes (e.g. Cohen's *d*, Pearson's *r*), indicating how they were calculated |

*Our web collection on statistics for biologists contains articles on many of the points above.*

## Software and code

Policy information about availability of computer code

| Data collection | For behavioral experiments, female aggression experiments were recorded from above using a camera (USB 3.1 Blackfly S, Monochrome Camera; Point Gray, Richmond, Canada) with an 800 nm long pass filter (B and W filter; Schneider Optics, Hauppauge, NY) at 170 frames per second and 1024 × 1024 pixel resolution. Animals were monitored during behavioral analysis using previously described and published BIAS software and MATLAB R2019a scripts (MathWorks). Male courtship experiments and animal trajectories collected during imaging were performed as detailed in Hindmarsh Sten et al. (2021). |
|---|---|
| Data analysis | Behavioral and imaging experiments were analyzed using MATLAB R2019a (MathWorks) and electrophysiological recordings were analyzed using pClamp (Clampfit 11.3). MATLAB (MathWorks) and GrapPad Prism 10 were used for statistics and data visualization. |

For manuscripts utilizing custom algorithms or software that are central to the research but not yet described in published literature, software must be made available to editors and reviewers. We strongly encourage code deposition in a community repository (e.g. GitHub). See the Nature Portfolio guidelines for submitting code & software for further information.

## Data

Policy information about availability of data

All manuscripts must include a data availability statement. This statement should provide the following information, where applicable:
- Accession codes, unique identifiers, or web links for publicly available datasets
- A description of any restrictions on data availability
- For clinical datasets or third party data, please ensure that the statement adheres to our policy

> All data underlying figures is available as supplementary materials (see Source Data) and code is provided in figshare (DOI: 10.25378/janelia.26849083) and github (https://github.com/ceschretter/SocialState2024_Code). Raw data is available upon request from the corresponding author.

## Research involving human participants, their data, or biological material

Policy information about studies with human participants or human data. See also policy information about sex, gender (identity/presentation), and sexual orientation and race, ethnicity and racism.

| | |
|---|---|
| Reporting on sex and gender | N/A |
| Reporting on race, ethnicity, or other socially relevant groupings | N/A |
| Population characteristics | N/A |
| Recruitment | N/A |
| Ethics oversight | N/A |

Note that full information on the approval of the study protocol must also be provided in the manuscript.

# Field-specific reporting

Please select the one below that is the best fit for your research. If you are not sure, read the appropriate sections before making your selection.

☒ Life sciences          ☐ Behavioural & social sciences          ☐ Ecological, evolutionary & environmental sciences

For a reference copy of the document with all sections, see nature.com/documents/nr-reporting-summary-flat.pdf

# Life sciences study design

All studies must disclose on these points even when the disclosure is negative.

| | |
|---|---|
| Sample size | No statistical methods were used to pre-determine sample size. Sample size was based on previous literature in the field. |
| Data exclusions | For tethered courtship experiments, only experiments during which animals exhibited courtship towards the visual targets were included for analysis. For all other behavioral and imaging experiments, data was only excluded in the event of acquisition error or data corruption. In Extended Data Figure 5b and 7a – e, outlier samples with very low intensities or those whose intensity randomly fluctuated were excluded from the analysis. |
| Replication | All attempts at replication were successful. Biological replicates completed at separate times using different parental crosses were performed for each of the behavioral experiments. Behavioral data are representative of at least two independent biological repeats. All functional imaging and electrophysiological recordings were replicated across at least three separate animals. |
| Randomization | As controls were performed within animals, no randomization was performed. This was not relevant for this study as all measurements of behavior and cell activity were quantitative assessments with no treatment groups. |
| Blinding | Experimenters were not blinded as all data acquisition and analysis was automated. |

# Reporting for specific materials, systems and methods

We require information from authors about some types of materials, experimental systems and methods used in many studies. Here, indicate whether each material, system or method listed is relevant to your study. If you are not sure if a list item applies to your research, read the appropriate section before selecting a response.

## Materials & experimental systems

| n/a | Involved in the study |
|-----|------------------------|
| ☐ | ☒ Antibodies |
| ☒ | ☐ Eukaryotic cell lines |
| ☒ | ☐ Palaeontology and archaeology |
| ☐ | ☒ Animals and other organisms |
| ☒ | ☐ Clinical data |
| ☒ | ☐ Dual use research of concern |
| ☒ | ☐ Plants |

## Methods

| n/a | Involved in the study |
|-----|------------------------|
| ☒ | ☐ ChIP-seq |
| ☒ | ☐ Flow cytometry |
| ☒ | ☐ MRI-based neuroimaging |

## Antibodies

| | |
|---|---|
| Antibodies used | Primary antibodies used were mouse nc82 (Developmental Studies Hybridoma Bank, nc82-s) and rabbit polyclonal α-GFP (Life Technologies, A11122). Secondary antibodies used were AF568 Goat α-Mouse (Life Technologies, A11031) and AF488 Goat α-Rabbit (Life Technologies, A11034). |
| Validation | All antibodies for this study have been used and validated previously (https://www.janelia.org/project-team/flylight/protocols). |

## Animals and other research organisms

Policy information about studies involving animals; ARRIVE guidelines recommended for reporting animal research, and Sex and Gender in Research

| | |
|---|---|
| Laboratory animals | All flies used in behavioral and functional analysis were between 3 – 10 days post eclosion. Images of brains are males and females and are indicated on the figure and figure legend. Please refer to the methods and resources table for additional description of research animals. |
| Wild animals | No wild animals were used in this study. |
| Reporting on sex | Males and females were used throughout the study and carefully considered during the study design. For experiments examining female aggression, only females were used as the cell types being studied differ between males and females as does the line expression. For experiments examining male courtship behavior, males and females were used. Behavioral analysis of the male was performed to compare across sexes. Sex in this study is reported in each figure legend as well as throughout the text. |
| Field-collected samples | No field-collected animals were used in this study. |
| Ethics oversight | No ethical approval was required as all experiments were performed in Drosophila melanogaster. |

Note that full information on the approval of the study protocol must also be provided in the manuscript.

## Plants

| | |
|---|---|
| Seed stocks | *Report on the source of all seed stocks or other plant material used. If applicable, state the seed stock centre and catalogue number. If plant specimens were collected from the field, describe the collection location, date and sampling procedures.* |
| Novel plant genotypes | *Describe the methods by which all novel plant genotypes were produced. This includes those generated by transgenic approaches, gene editing, chemical/radiation-based mutagenesis and hybridization. For transgenic lines, describe the transformation method, the number of independent lines analyzed and the generation upon which experiments were performed. For gene-edited lines, describe the editor used, the endogenous sequence targeted for editing, the targeting guide RNA sequence (if applicable) and how the editor was applied.* |
| Authentication | *Describe any authentication procedures for each seed stock used or novel genotype generated. Describe any experiments used to assess the effect of a mutation and, where applicable, how potential secondary effects (e.g. second site T-DNA insertions, mosiacism, off-target gene editing) were examined.* |

