## [Peer Review File · Nature]

Manuscript Title: Social state alters vision using three circuit mechanisms in *Drosophila*

Reviewer Comments & Author Rebuttals

Reviewer Reports on the Initial Version:

Referees' comments:

Referee #1 (Remarks to the Author):

In this ms, Vanessa Ruta, Gerald Rubin and colleagues investigate the neural mechanisms underlying aggressive behavior in fruit flies. They find three circuit motifs by which a central neuron, called aIPg, when active, affects and routes the activity of visual interneurons to selectively respond to images of conspecifics and initiates aggressive behavior: selective amplification of output, dendritic disinhibition and toggle switch between different visual interneurons. The ms represents an impressive body of experiments, including behavior, physiology and connectomics. As far as I can see, all experiments are thoroughly conducted, conclusive and support the above statements. In general, thus, I find this work highly interesting, not just for Drosophilists but for a general neuroscience audience as well. Nevertheless, there is still some work to do which I specify below.

Major comments:

1. Contribution of different LC neuron types to female-female aggression: Several LC neuron types detect discrete objects with angular sizes within the range a fly can subtend on the retina of an observer fly, or in the author's words 'fly-sized moving objects'. LC11, LC10a, LC12, LC21, LC18, as well as LC9 (as per this manuscript) and counting. It is indeed puzzling that most of them are dispensable for female tracking (as shown by Cowley, et al, 2023). Although Bidaye, et al, 2020 reported a decrease in following in males with silenced LC11 neurons, there are at least two other papers using single-pair courtship assays that show LC11 neurons are dispensable for tracking: Ribeiro et al 2018 reported normal levels of female tracking upon silencing LC11 neurons in a Supplementary Figure and Cowley et al 2023 show a median distance male-female well within control levels in Figure 1. In addition, Tanaka et al, 2020 reported LC11 neurons are both required and sufficient for slowing down. In all likelihood Bidaye et al, 2020 mixed up the line for LC11 (OL15B) with the line for LC10a (OL19B), both from Nern, Wu, et al, 2016. The authors could test whether LC11 and LC9 are required for female-female aggression to clear this matter.

2. Presentation of the data and structure of the ms: I found it extremely difficult to extract the relevant information from the ms. The fact that so many different cell types appear makes it already hard for any reader outside the field, but this is unavoidable. Therefore, an extra effort for clarity is required in the structure of the ms. Let me give you a few examples: Figure 1a represents a nice graphical overview of the problem (to the left) and the three circuit motifs (to the right). But 1b-g focus on the first mechanism exclusively. Figure 1b, for example, is an almost incomprehensible summary of connectomics data about postsynaptic neurons receiving input both from various subtypes of LC10 and aIPg. Why not leave it there and have Fig1a, together with the circuit scheme of Figure 6 (this summary diagram comes rather late), as Figure 1 alone? Figure 2-4 each would then focus on one of the three mechanisms. These three data figures could follow a common rule and include a) a picture of the respective sub-circuit, b) the behavioral effect of blocking the respective interneuron (like Figure 2c for mechanism 1, figure 3b for mechanism 2 and figure 5a for mechanism 3), and c) the relevant physiology (calcium imaging of postsynaptic neuron to presynaptic stimulation, receptive field tuning etc). Along the same line, the numbering of figures doesn't seem

to follow any rule: Wouldn't it be appropriate to have Figure 2 followed by Figure 2, extended data a,b,c, before moving to Figure 3? As an example, in line 160, Figure 6 and extended data figure 9 appear after Figure 3. The sequence of the figures themselves is also confusing with Extended Data Figure 3 (line 604) appearing before Figure 3 (line 616).

Other comments and questions:

1. LC10-group neurons are a diverse group of neuron types, that grossly share the neuropiles where both dendrites and axon terminals reside. Morphologically they are quite different (Nern, Wu, 2016; Seung 2023), display different transcriptomes (Davis, 2020) and appear to mediate different behaviors (Ribeiro, 2018, Hindmarsh-Sten2021, and this manuscript!) – or at least of all LC10s, only LC10a mediate female tracking or are required for female-female aggression. LC10 neuron types thus are not subtypes. As a comparison: T4 or T5 subtypes (a,b,c,d) have very similar morphologies, all are motion detectors, and differ only in the orientation of the dendrites and the direction of motion each subtype is sensitive to. Comparing with T4 and T5 subtypes, LC10-group neurons can't be called subtypes. In line with this, the term 'LC10' appears isolated several times in the text. Since there is no 'LC10' neuron type, it is important to refer to LC10-group neuron types more explicitly. If 'LC10' refers to a specific LC10 type, then indicate the type. When 'LC10' refers to all LC10-types, either use LC10s or LC10-group neuron types or LC10-types. Keeping 'LC10' only generates more confusion in the literature. The lobula plate tangential cells (many neuron types) are also never referred to as LPTC (or LPT), but as LPTCs or LPTs instead.

2. The aggression level upon stimulation of aIPg seems to vary quite substantially, like almost 70% in Figure 2c left panel, 80% in the same figure, right panel, 60% in Figure 5a, down to less than 40% in Figure 5a, right panel. Do the authors have any explanation for that large variability?

3. The data in Figure 2a, showing the preference of LC10a for object width and height which peaks at about 15 degree, seem to disagree with the one of extended data figure 2d, where the maximum responses of LC10a (?) has a maximum at around 9 degree. In the legend to extended data figure 2 the authors write that the stimuli moved at a slowed speed of 50 deg/s, but was the speed used in Figure 2a?

4. Why do the authors, when showing a response of a neuron (Tutu1 or 2) to stimulation of aIPg, write: 'Excitatory responses ... before, during, and following a 2 ms stimulation ...', instead of: 'Excitatory responses ... to a 2 ms stimulation ...'

5. What is iLexA in the genetic access to aIPg? In this line: 72C11-iLexA.

6. How many females are placed in the arena of 53.3 mm for each assay? The measure for aggression behavior 'Avg. Aggression (%)' and 'Aggressive behaviors (%)' appear often, for example in Extended Data Figure 4. What is this measure exactly? Is it the percentage of time in one assay where any female shows any action related to aggression? Or is this measure computed per fly?

7. Most of the experiments in this manuscript use artificial activation of aIPg neurons to elicit aggression. Female-female aggression also occurs in the absence of activation of aIPg neurons (Schretter, et al, 2020). If this baseline female-female aggression were to be measured the same way as it is measure in aIPg activation, with the Avg. Aggression (%) instead of total amount of time (in Schretter, et al, 2020), how much would it be? Is the panel g in Extended Data Figure 2 a representative estimation of such a baseline level? It is very low.

8. Where is the data supporting the circuit diagram shown in Figure 3b? It would be great to have

more detail about these lobula interneurons. Aren't they present in the Hemibrain or the Flywire whole brain? A diagram of the IB112 in the lobula would be nice – like the one in Figure 3a for IB112 with aIPg.

9. The 'Angle occluded by nearest fly' is the angular size of the nearest fly? How is this angle computed?

10. What neuron replaces aIPg in males? Perhaps the authors could speculate about this.

Referee #2 (Remarks to the Author):

Schretter...Rubin 2024 (Nature) Review

This fantastic article by Catherine Schretter and colleagues takes full advantage of the *Drosophila* toolkit to explore circuit mechanisms that could underlie the arousal-based gating of visual information flow in LC neurons. The work begins with a connectome analysis of LC cell types, their downstream targets, and aIPg, which the authors have previously shown to be involved in female aggression behavior. The authors highlight three circuit motifs that could underlie aIPg's influence, including (1) convergence with sensory information, (2) dendritic disinhibition, and (3) a "toggle switch" that presides over mutual inhibition. The remainder of the paper is a methodological tour-de-force to assess the functional relevance of these circuit architectures, including thermogenetics, optogenetics, electrophysiology, calcium imaging, split-GAL4 genetic reagent creation, free behavior assays, VR stimulus presentation, and an impressive number of experiments that combine two or more of these techniques. The authors are (mostly, see below) able to convincingly demonstrate that each of these three circuit mechanisms are capable of modulating at least LC10a activity.

Overall, the work reported here is substantial, and the results will draw interest from a number of disciplines. In fly neuroscience, connectome analyses are particularly "hot" right now with the release of the FAFB dataset for free community use, and a number of connectome-driven circuit-cracking papers are likely to be on the horizon. While papers of a similar flavor have been popping up in larvae for quite some time (see for example Jovanic et al.), the present work is among the vanguard of a new, larger wave of connectome excitement. More generally, context-dependent modulation of neuronal activity seems to be another popular area of study. This paper expertly leverages fly-exclusive resources to address fundamental circuit architectures that can facilitate such modulation. For this reason, I believe this paper will be of considerable interest to a wide audience.

To my mind, the most significant conceptual advances here are at the circuit implementation level. Excitatory convergence, dendritic disinhibition, axo-axonic "shunting" inhibition, and mutual inhibition are not new concepts, as the authors report. The identification of specific neurons performing these computations in an otherwise well-described sensorimotor circuit is, I believe, what makes this paper exceptional. This paper offers the first description of a computational role for a visual centrifugal neuron, and, to my knowledge, is the first to report a toggle-like switch neuron that presides over mutual inhibition circuitry. For these reasons, I believe this paper represents a significant advance.

I do, however, have some concerns (related to the most significant conclusions of the manuscript) that should be addressed.

1. There is some conflation of LC10a's role in male courtship and female aggression that occurs throughout the manuscript. Without some additional experiments, I believe the authors should adjust their language regarding a common circuit architecture mediating sensory gating in diverse behavioral contexts.

a. The most crucial case involves the gating of LC10a visual activity, which (to my knowledge) has not been directly observed in females. The authors record robust visual responses in a non-aggressive state in Figure 2a, which actually makes a strong case that the arousal-based gating seen in males is actually not present in females. Repeating the experiments of Figure 2a in a thermogenetically aIPg-activated context would resolve this question. In the absence of this important experiment, the authors must clarify their reasoning, as this disrupts the logical chain from circuit mechanism to visual gating to behavioral context.

b. The authors' conclusion that selective modulation of LC10 subtypes shapes male courtship (Figure 5) is currently not well supported. The TuTu recording experiments, while elegant and informative, do not establish a causal role. And the behavioral activation experiments in Extended Data Figure 7g are missing some controls and could use some additional analysis. Metrics such as copulation success rate and time to copulation, which are often used to describe the mating effects of visually impairing males, would strengthen the conclusion. Further, the empty-gal4 negative control (see Tadres et al.), found in other experiments throughout the paper, is conspicuously lacking here.

2. The functional connectivity experiments using Chrimson/GtACR need empty-gal4 controls. One control for each imaged/patched downstream neuron would suffice, to demonstrate that the observed responses are not due to nonspecific expression leak or a photoelectric/intrinsic photosensitivity effect. See Tadres et al. In the absence of these controls, I think the authors need to be careful about over-stating the functionality of connections suggested by their connectome analysis. I believe the control is especially critical for the dendritic disinhibition mechanism (mechanism 2). There are hints that nonspecific mechanisms might be relevant here, given the order of magnitude more stimulation required to minimally activate IB112 compared to TuTuAs.

3. A core discussion topic is that the toggle switch mechanism described by the authors would act to better align information flow in the AOTu to the stimulus statistics of flies engaging in aggressive behavior. If my reading of the data is correct, the distribution of retinal object size in aggressive females and the tuning of LC10a are only glancingly aligned, with the peak of the retinal size distribution lying on the edge of the LC10a tuning curve. As such, the authors should soften their statements about LC10a's suitability for encoding visual features during aggressive bouts. Additionally, it's not clear what is known about the tuning of LC10c. It might make sense to perform another set of recording experiments (like the ones in Figure 2a) for LC10c. Then, the authors can show that aIPg activation shifts visual information flow from LC10c tuning (presumably even less well aligned with stimulus statistics) to LC10a tuning (as already described).

Beyond these concerns, I have a few data presentation/statistics issues. First, I found the presentation method for genotypes/genetic controls quite difficult to interpret. I believe a general revision of bar plot labeling is called for here. A reader would not quickly grok what is actually being tested in Fig 2c, for example. Across figures, "positive/negative control" can mean different things. Filling in a bubble with "aIPg active" in an experiment where GtACR silencing occurs is misleading. And so on. Rather than using these simple word models for each experiment, I believe the standard approach of showing which alleles and/or manipulations are present in each case would be much clearer, and there would be less risk of misinterpretation.

Finally, there is a good deal of variance in the group sizes used in a number of statistical tests. Most worrying are the TuTuA experiments in Figure 5a, with group sizes ranging from 6 to 20. Given the trends towards effects in the empty-gal4 controls that can be clearly seen in Extended Data Figure 7b&c, dramatically different group sizes are problematic. I believe the authors should collect additional data to bring the group sizes into better alignment, so that readers can be more confident in the paper's conclusions.

Minor Comments:

1. The authors refer to the sensory gating here as “visual attention” a few times, and I’m not confident this term applies. “Visual attention” typically refers to gain enhancements in a spatial subset of the visual field, whereas this paper describes gain enhancements to particular visual features, anywhere in the visual field. I believe “context-dependent gating of visual features” is more appropriate than “visual attention.”
2. I believe a citation of Turner et al. is more than warranted, as it is one of the few prior examples of LC modulation by behavioral context.
3. In Figure 2 and Extended Data Figure 2, there is a clear effect on aggression mediated by LC10b/c neurons. However, the particular way the data are summarized obscures this effect in the main Figure. While the author’s wording is technically correct (“LC10a, but not LC10b/c, resulted in a sustained decrease...” [emphasis is mine]), I don’t think we are doing justice to the complexity of the data by pushing this aside. The sign of the effect mediated by LC10b/c seems like it might be consistent with the 10a vs. 10c toggle switch mechanism. Could this result be folded into the narrative?
4. In Figure 4c-d, I am having a hard time assessing the degree to which the TuTuA inputs are axo-axonic. The synapse sites don’t appear particularly close to the yellow presynaptic terminals. Is this because we are only looking at a portion of the larger arbor? If so, having a full skeleton present that also shows postsynaptic sites, and then doing the existing images as zoomed-in insets, might help to clarify that these are indeed close to specifically presynaptic sites.
5. In Figure 4e-f, a non-nAChR-mediated residual is present, but I was unable to find a discussion of this in the text. What do you think is the source of this input? Is this a problem for your model, and if not, why?
6. In Figure 5b, adding SMP054 and LC10a labels to the schematic will help improve clarity and consistency.
7. The scale of the Y-axis in Figure 5b is dramatically different from every other plot of this kind in the manuscript. I know this must be the case in order to properly illustrate the effect, but I wonder if there is something that can be done to make the plot a little more visually different from the others, to cue readers into this fact at a glance? Perhaps not having it horizontally aligned with the plots in panel a would be enough.
8. In Figure 6, I believe the toggle-switch mechanism would be easier to appreciate if you split the LC10 population into LC10a and LC10c. Then this would be a great way to see all three mechanisms in the same circuit diagram, and how they relate to the proposed output pathway in the AOTu.
9. Lines 121-122: “limited subset of LC neurons tuned to fly-sized moving objects” – my understanding from Klapoetke et al. and Turner et al. is that a large fraction of recorded LC/LPLC types respond to small moving objects. And as the authors explain, with varying distance, the size of a “fly-sized object” will change. As such, I think the “limited subset” phrase should be removed. Additionally, I think it would be fair to add some graphics or language about the aIPg connectivity to other moving spot-tuned cell types, here and/or in Extended Data Figure 9.
10. Lines 207-208: unless I’m missing something in the circuit diagram, the idea that this organization would primarily mediate feedback inhibition seems a bit weak, given the axo-axonic

connectivity pattern.

11. Lines 208-211: this sentence is a bit opaque, it's not clear what interrogations are being proposed, nor is it clear why they can't currently be done. Does this point need to be made?

12. Line 249: I'm not sure it's fair to say that baseline physiological properties of TuTu neurons are the same across sexes based only on the data shown in Extended Data Figure 8. The action potential shape is shown to be different, and there seems to be significant differences in burstiness/rhythmicity that may signal differential expression of ion channels. Without more analyses and/or experiments, I recommend softening this language.

13. Line 625: I believe there is a typo in a number of genotypes written in legends. This is one example. Switching to the conventional genotype-based bar plot labels (see above) will obviate the need for these big genotype lists.

14. Line 720: Should add "optogenetic" here (in reference to the stimulus). Related, why is the activation so long here compared to the patching experiments? Related to major concern 2, above.

References:

Hindmarsh Sten T, Li R, Otopalik A, Ruta V. Nature 2021

Jovanic T, Schneider-Mizell CM, Shao M, Masson JB, Denisov G, Fetter RD, Mensh BD, Truman JW, Cardona A, Zlatić M. Cell 2016

Klapoetke NC, Nern A, Rogers EM, Rubin GM, Reiser MB, Card GM. Neuron 2022

Tadres D, Shiozaki HM, Tastekin I, Stern DL, Louis M. bioRxiv 2022

Turner MH, Krieger A, Pang MM, Clandinin TR. Elife 2022

Referee #3 (Remarks to the Author):

The circuits and mechanisms of early visual processing is rather well understood across various animal clades, particularly in insects (or arthropods generally) and vertebrates. However, while a wealth of studies in mammals (particularly based on electrophysiological approaches) has identified strikingly complex concepts of visual representations in deeper brain regions and to some degree the circuit organization the corresponding counterparts are less well understood in non-vertebrates.

The current manuscript provides an elegant and powerful example of visual processing of potentially complex type of cues using the fruit fly as model. It is a beautiful example of how connectomics may be employed together with high-precision genetic tools and physiological and behavioral analyses to decipher how higher brain circuits are able to achieve surprisingly advanced functions. It hereby focusses on the state-dependent change in visual processing in female flies.

There are a series of intriguing and highly relevant points in the study. Above all, it shows that an internal state such as aggression can alter how visual information is processed; I find this particularly relevant since aggression is by itself a rather complex state (as compared to starvation). Secondly, the authors layout three different state-dependent circuit motifs in a rather great vigor.

I do not have any major criticism at the work, however there are a few points, that may benefit from

clarification or further explanation.

While connectomic data are powerful to depict (and to some degree prove) the existence of chemical synapses between certain neurons, there are several limitations to the analyses. One particularly relevant limitation to the current work is that the connections can merely be defined by the highest or most abundant synapses, however the actual synaptic weight is much more critical than synapse number (e.g. 10 strong synapses may be overcoming 100 weak ones). Do the authors have any evidence in this specific circuit for a correlation of this? Similarly, while they show axonal termini, do they have evidence that the actual synapse number in an average female fly is indeed similar to the one in the connectome (which represents a specific snapshot of one individual)?

Do the authors have any evidence for the molecular input pathways of the “aggression state”? The behavioral paradigm is only briefly mentioned and since it’s a core aspect of the manuscript may deserve better explanation. As it reads now it’s rather a miraculous state-switch, without much insight into the cues and input pathways.

While the connectome is based on a female individual and much of the current work is focused on female aggression, the authors also provide evidence for the presence of the same circuit in male (courtship). This is particularly intriguing, since the recognition of a conspecific fly (independent of male/female) elicits different responses in both sexes. Is it to some degree or approximation possible to explain this in the current knowledge of male/female specific circuits and the existence of split *gal4* lines and sex-specific neural lines?

Author Rebuttals to Initial Comments:

Schretter et al., 2024
Response to Reviewers

We would like to thank the reviewers for their comments. Addressing the issues they raised allowed us to significantly improve the manuscript. More specifically, we have reorganized the manuscript, figures, added additional data and analyses, and made text changes as requested. A point-by-point response to the reviewers' comments is provided below.

We have included two versions of the manuscript, one unmarked and one in which the major changes are tracked. Line numbers listed below refer to the unmarked manuscript as track changes produced line number errors.

Referee #1 (Remarks to the Author):

In this ms, Vanessa Ruta, Gerald Rubin and colleagues investigate the neural mechanisms underlying aggressive behavior in fruit flies. They find three circuit motifs by which a central neuron, called aIPg, when active, affects and routes the activity of visual interneurons to selectively respond to images of conspecifics and initiates aggressive behavior: selective amplification of output, dendritic disinhibition and toggle switch between different visual interneurons. The ms represents an impressive body of experiments, including behavior, physiology and connectomics. As far as I can see, all experiments are thoroughly conducted, conclusive and support the above statements. In general, thus, I find this work highly interesting, not just for Drosophilists but for a general neuroscience audience as well. Nevertheless, there is still some work to do which I specify below.

Major comments:

1. Contribution of different LC neuron types to female-female aggression: Several LC neuron types detect discrete objects with angular sizes within the range a fly can subtend on the retina of an observer fly, or in the author's words 'fly-sized moving objects'. LC11, LC10a, LC12, LC21, LC18, as well as LC9 (as per this manuscript) and counting. It is indeed puzzling that most of them are dispensable for female tracking (as shown by Cowley, et al, 2023). Although Bidaye, et al, 2020 reported a decrease in following in males with silenced LC11 neurons, there are at least two other papers using single-pair courtship assays that show LC11 neurons are dispensable for tracking: Ribeiro et al 2018 reported normal levels of female tracking upon silencing LC11 neurons in a Supplementary Figure and Cowley et al 2023 show a median distance male-female well within control levels in Figure 1. In addition, Tanaka et al, 2020 reported LC11 neurons are both required and sufficient for slowing down. In all likelihood, Bidaye et al, 2020 mixed up the line for LC11 (OL15B) with the line for LC10a

(OL19B), both from Nern, Wu, et al, 2016. The authors could test whether LC11 and LC9 are required for female-female aggression to clear this matter.

We performed the suggested experiments examining if other LCs are necessary for aIPg-induced female aggression. We tested the reviewer's suggestion by silencing LC9 and LC11 neurons and added LC15 as a control. LC15 displays similar responses to moving objects (Klapoetke et al., 2022) but lacks shared downstream targets from aIPg. We found no decrease in the average bouts of aggression during LC9, LC11, or LC15 inactivation (Extended Data Figure 3g). Notably, both the time course and average duration of aggression for LC9 showed an overall small decrease in the baseline aggression that did not rebound, similar in direction to the much larger effect we observed upon LC10a inactivation (Extended Data Figure 3f). However, additional connectomics analyses revealed that LC9 provides substantial dendro-dendritic input to LC10a (see pie chart in Figure 3b); therefore, any suppression of aggression arising from modulation of LC9 activity is confounded by these connections to LC10a. Upon LC11 optogenetic inactivation, we did observe a slight increase in the average percentage of female aggression (Extended Data Figure 3g). However, based on comparing aggression during the optogenetic stimulus on and off periods, we believe this is most likely due to an extended decrease in aggression during the stimulus off period as there was no increase in aggression above the negative control (empty > GtACR) during the stimulus on period (Extended Data Figure 3f). We have added this data to the revised manuscript (Extended Data Figure 3e – g) and refer to it in Lines 148 – 151: "Optogenetic inactivation of LC10a, but not LC10bc, LC9, or LC11, resulted in a sustained decrease in aggressive behaviors, including individual component features such as touch (Figure 3f and Extended Data Figure 3a – g)."

While this additional data suggests LC9 and LC11 might not contribute to the components of aIPg-mediated female aggression that we measure here, we cannot rule out that inactivating single LC types may be compensated by other types. Nevertheless, our new data further support that LC10a is distinct from these other LCs and its inactivation cannot be fully compensated by other cell types.

2. Presentation of the data and structure of the ms: I found it extremely difficult to extract the relevant information from the ms. The fact that so many different cell types appear makes it already hard for any reader outside the field, but this is unavoidable. Therefore, an extra effort for clarity is required in the structure of the ms. Let me give you a few examples: Figure 1a represents a nice graphical overview of the problem (to the left) and the three circuit motifs (to the right). But 1b-g focus on the first mechanism exclusively. Figure 1b, for example, is an almost incomprehensible summary of connectomics data about postsynaptic neurons receiving input both from various subtypes of LC10 and aIPg. Why not leave it there and have Fig1a, together with the circuit scheme of Figure 6 (this summary diagram comes rather late), as Figure 1 alone? Figure 2-4 each would then focus on one of the three mechanisms. These three data figures could follow a common rule and include a) a picture of the respective sub-circuit, b) the behavioral effect of blocking the respective interneuron (like Figure 2c for mechanism 1, figure 3b for mechanism 2 and figure 5 a for mechanism 3), and c) the

relevant physiology (calcium imaging of postsynaptic neuron to presynaptic stimulation, receptive field tuning etc). Along the same line, the numbering of figures doesn't seem to follow any rule: Wouldn't it be appropriate to have Figure 2 followed by Figure 2, extended data a,b,c, before moving to Figure 3? As an example, in line 160, Figure 6 and extended data figure 9 appear after Figure 3. The sequence of the figures themselves is also confusing with Extended Data Figure 3 (line 604) appearing before Figure 3 (line 616).

We thank the reviewer for this suggestion and have restructured the manuscript and figures accordingly. For clarity, we combined the original Figure 1a and Figure 6 into the updated Figure 1 and separated the original Figure 1b – g into the new Figure 2. We also separated the original Extended Data Figure 2 into two figures (updated Extended Data Figure 2 and 3) to clarify which data corresponded to LC feature detection versus involvement in female aggressive behavior.

We agree with the reviewer that the numbering of extended data figures can be confusing; however, this was done to conform to the policies of Nature on figure numbering.

Other comments and questions:

1. LC10-group neurons are a diverse group of neuron types, that grossly share the neuropiles where both dendrites and axon terminals reside. Morphologically they are quite different (Nern, Wu, 2016; Seung 2023), display different transcriptomes (Davis, 2020) and appear to mediate different behaviors (Ribeiro, 2018, Hindmarsh-Sten2021, and this manuscript!) – or at least of all LC10s, only LC10a mediate female tracking or are required for female-female aggression. LC10 neuron types thus are not subtypes. As a comparison: T4 or T5 subtypes (a,b,c,d) have very similar morphologies, all are motion detectors, and differ only in the orientation of the dendrites and the direction of motion each subtype is sensitive to. Comparing with T4 and T5 subtypes, LC10-group neurons can't be called subtypes. In line with this, the term 'LC10' appears isolated several times in the text. Since there is no 'LC10' neuron type, it is important to refer to LC10-group neuron types more explicitly. If 'LC10' refers to a specific LC10 type, then indicate the type. When 'LC10' refers to all LC10-types, either use LC10s or LC10-group neuron types or LC10-types. Keeping 'LC10' only generates more confusion in the literature. The lobula plate tangential cells (many neuron types) are also never referred to as LPTC (or LPT), but as LPTCs or LPTs instead.

We agree with the reviewer and have changed all references to all LC10s to "LC10s" or "LC10-group cell types."

2. The aggression level upon stimulation of aIPg seems to vary quite substantially, like almost 70% in Figure 2c left panel, 80% in the same figure, right panel, 60% in Figure 5a, down to less than 40% in Figure 5a, right panel. Do the authors have any explanation for that large variability?

The variability in female aggression we report across experiments is due to systematic variation in baseline aggression in different genotypes and interindividual variability within a genotype and the fact that averaging is performed across individuals. Each dot in the main figure represents one experiment containing 5 – 8 flies within the arena. The average aggression is calculated across all flies in each experiment. In addition to providing descriptions of our analysis in the methods section and figure legends, for clarity we have now also added a diagram in the main figure schematizing how the data from a representative experiment was processed to generate a single pair of data points comparing aggression during the baseline and optogenetic stimulation (Figure 3e).

As shown in Figure 3f (original Figure 2c), there is variation in the average aggression during the baseline periods. The variation in baseline aggression is due to many different factors, including the genotype, which is why comparisons are only made within a genotype rather than across genotypes. Some variation in the same genotype across figures (e.g. *alPg>TrpA empty >GtACR*) may also arise from the way that the baseline period was defined. For example, in Figure 3f, the baseline was calculated from the 15 s before the first stimulus period, as stated in the figure legend, to account for the fact that the *LC10a > GtACR* group did not return to baseline between stimulation epochs. However, in Figure 4c and Extended Data Figure 3g, the baseline aggression was calculated by taking the average in the 30 sec prior to all three stimulation epochs. These calculations are defined in the figure legends and the description of the analysis performed for each figure in Supplementary File 1.

We include the time courses for all data in the Extended Data Figures as a reference for clarity and transparency of all the data that was used to generate the average values shown in the main figures.

3. The data in Figure 2a, showing the preference of LC10a for object width and height which peaks at about 15 degree, seem to disagree with the one of extended data figure 2d, where the maximum responses of LC10a (?) has a maximum at around 9 degree. In the legend to extended data figure 2 the authors write that the stimuli moved at a slowed speed of 50 deg/s, but was the speed used in Figure 2a?

We thank the reviewer for this point and have added that the stimulus speed in Figure 3a ('100 deg/s') (original Figure 2a). As the reviewer suggests, this is indeed the reason for the difference in the responses between Figure 3a and Extended Data Figure 2d.

4. Why do the authors, when showing a response of a neuron (Tutu1 or 2) to stimulation of *alPg*, write: 'Excitatory responses ... before, during, and following a 2 ms stimulation ...', instead of: 'Excitatory responses ... to a 2 ms stimulation ...'

We have changed the wording as suggested in the figure legends for Figure 5e – f and Extended Data Figure 7g – j.

5. What is *iLexA* in the genetic access to *alPg*? In this line: *72C11-iLexA*.

We have now added the following clarification in the methods. “iLexA stands for improved LexA, in which additional activating domains were added the transcriptional activator to enhance expression (Chavez et al., 2016). For the aIPg-iLexA line, the pBPnIsLexA::p65::GADUw vector was used for cloning as in Chiu et al., 2021.”

6. How many females are placed in the arena of 53.3 mm for each assay? The measure for aggression behavior ‘Avg. Aggression (%)’ and ‘Aggressive behaviors (%)’ appear often, for example in Extended Data Figure 4. What is this measure exactly? Is it the percentage of time in one assay where any female shows any action related to aggression? Or is this measure computed per fly?

As detailed in the Methods, 5 – 8 females were monitored in the arena for each experiment. Since aggression is inherently the interaction of more than one fly, we cannot disassociate the behavior of one fly from that of the others. Aggression was therefore computed on a per arena basis rather than a per fly basis (Robie et al., 2017, Schretter et al., 2020). Flies were tracked using Caltech FlyTracker followed by automated classification of behavior with JAABA classifiers. The aggression classifier was defined based on previous literature (Schretter et al., 2020, Nilsen et al., 2004) and encompassed both fencing and head butting behaviors. The percent aggression is the sum of the aggression scores for all the trajectories in which flies were performing aggression divided by the total number of flies in the arena times 100. The time series graphs plot the mean calculated over 0.35 s (60-frame) bins. Unless otherwise stated, the average aggression is calculated by averaging over the three optogenetic stimulus off periods or three optogenetic stimulus (stimulus on) periods. We have also now added the following text explaining these calculations (Lines 142 – 146): “To assess the involvement of specific cell types in aIPg-mediated aggression, we recorded, tracked, classified, and analyzed the aggressive behavior of groups of female flies in an arena as diagrammed in Figure 3e. We quantified the average percent of trajectories in which flies were performing aggression in each such experiment, which we represented as a single point in our graphs and analyzed across approximately 20 experiments per condition (Figure 3e).” as well as the schematic in Figure 3e to clarify these metrics.

7. Most of the experiments in this manuscript use artificial activation of aIPg neurons to elicit aggression. Female-female aggression also occurs in the absence of activation of aIPg neurons (Schretter, et al, 2020). If this baseline female-female aggression were to be measured the same way as it is measure in aIPg activation, with the Avg. Aggression (%) instead of total amount of time (in Schretter, et al, 2020), how much would it be? Is the panel g in Extended Data Figure 2 a representative estimation of such a baseline level? It is very low.

Like male courtship, female aggression is highly context dependent (Schretter et al., 2020, Bath et al., 2018, Bath et al., 2017, Ueda and Kidokoro, 2002). In the conditions used for this paper, the average baseline female aggression is 2.0% (compared to 66.3% during TrpA stimulation for the aIPg > TrpA empty > GtACR condition), which is comparable to the levels of aggression reported in our previous work (Schretter et al., 2020, Figure 1f).

The focus of our work was to examine the mechanisms by which aIPg-mediated aggression occurs; therefore, we activated aIPg to eliminate the effects from components upstream of aIPg. Although most experiments were performed with aIPg activation to promote aggression, we would like to note that our experiments on the TuTuA-mediated switch in Figure 6b suggest its involvement in wild type aggression since inactivation of TuTuA_2 alone increased aggression when aIPg was not exogenously activated.

8. Where is the data supporting the circuit diagram shown in Figure 3b? It would be great to have more detail about these lobula interneurons. Aren't they present in the Hemibrain or the Flywire whole brain? A diagram of the IB112 in the lobula would be nice – like the one in Figure 3a for IB112 with aIPg.

We have added this detail in the new Figure 4b and its corresponding figure legend (which has replaced Figure 3a in the original manuscript). Briefly, we performed additional analyses of the male and female optic lobe using the FlyWire and the newly released male optic lobe datasets. As reported in the revised text, we found that the lobula interneuron Li22 is IB112's top downstream target. Li22 cells receive 5.7% of their input from IB112 in the male optic lobe and 7.1% in the female. Li22 provides 3.5% of LC10a, 1.1% of LC10b, 5.1% of LC10c and 5.9% of LC10d dendritic inputs in the male, and 3.5%, 0.8%, 4.9% and 5.3% in the female, respectively.

9. The 'Angle occluded by nearest fly' is the angular size of the nearest fly? How is this angle computed?

Yes, we computed angle occluded by nearest fly using the `anglesub_perframe` feature as calculated in Robie et al., 2017. Briefly, this is defined as the “maximum total angle of the animal's field of view occluded by another animal... [and it is] computed by finding the two lines tangent to the fit ellipse that intersect at the nose-point of this fly and measuring the angle between them.” We also added this description to the Methods section (Lines 896 – 898).

10. What neuron replaces aIPg in males? Perhaps the authors could speculate about this.

The aIPg lineage expresses the sexually-dimorphic transcription factor, *fruitless* (*fru*) and generates different cells across sexes (Cachero et al., 2010). Other examples of sexually-dimorphic populations arising from the same lineage exist—for example, the pC1 lineage includes P1 neurons in males and pC1d/e neurons in females, which both are involved in promoting aggression. It is therefore conceivable that aIPg neurons in the male could be performing a similar role to the female aIPg neurons and serve to gate the visuomotor circuits involved in aggression. In the absence of a completed male connectome, we do not know which aIPg subtypes are involved or whether there are also P1 cell types that might connect directly to LC10a. Therefore, we believe

speculation about the exact role of male aIPg neurons is best left until after the male central brain connectome becomes available.

Referee #2 (Remarks to the Author):

Schretter...Rubin 2024 (Nature) Review

This fantastic article by Catherine Schretter and colleagues takes full advantage of the *Drosophila* toolkit to explore circuit mechanisms that could underlie the arousal-based gating of visual information flow in LC neurons. The work begins with a connectome analysis of LC cell types, their downstream targets, and aIPg, which the authors have previously shown to be involved in female aggression behavior. The authors highlight three circuit motifs that could underlie aIPg's influence, including (1) convergence with sensory information, (2) dendritic disinhibition, and (3) a "toggle switch" that presides over mutual inhibition. The remainder of the paper is a methodological tour-de-force to assess the functional relevance of these circuit architectures, including thermogenetics, optogenetics, electrophysiology, calcium imaging, split-GAL4 genetic reagent creation, free behavior assays, VR stimulus presentation, and an impressive number of experiments that combine two or more of these techniques. The authors are (mostly, see below) able to convincingly demonstrate that each of these three circuit mechanisms are capable of modulating at least LC10a activity.

Overall, the work reported here is substantial, and the results will draw interest from a number of disciplines. In fly neuroscience, connectome analyses are particularly "hot" right now with the release of the FAFB dataset for free community use, and a number of connectome-driven circuit-cracking papers are likely to be on the horizon. While papers of a similar flavor have been popping up in larvae for quite some time (see for example Jovanic et al.), the present work is among the vanguard of a new, larger wave of connectome excitement. More generally, context-dependent modulation of neuronal activity seems to be another popular area of study. This paper expertly leverages fly-exclusive resources to address fundamental circuit architectures that can facilitate such modulation. For this reason, I believe this paper will be of considerable interest to a wide audience.

To my mind, the most significant conceptual advances here are at the circuit implementation level. Excitatory convergence, dendritic disinhibition, axo-axonic "shunting" inhibition, and mutual inhibition are not new concepts, as the authors report. The identification of specific neurons performing these computations in an otherwise well-described sensorimotor circuit is, I believe, what makes this paper exceptional. This paper offers the first description of a computational role for a visual centrifugal neuron, and, to my knowledge, is the first to report a toggle-like switch neuron that presides over mutual inhibition circuitry. For these reasons, I believe this paper represents a significant advance.

I do, however, have some concerns (related to the most significant conclusions of the

manuscript) that should be addressed.

1. There is some conflation of LC10a's role in male courtship and female aggression that occurs throughout the manuscript. Without some additional experiments, I believe the authors should adjust their language regarding a common circuit architecture mediating sensory gating in diverse behavioral contexts.

We agree with the reviewer that our data primarily analyzes and supports the circuit architecture underlying female aggression. Based on past work on male courtship and our data in Figure 6 (original Figure 5), we believe that this architecture is likely more generalizable and may play a similar role whenever flies are interacting with other flies. However, in the absence of the male connectome, we agree that we cannot conclude that the circuit mechanisms are the same.

We have therefore both modified the concluding sentence of the LC10 section to (Lines 153 – 154, “Collectively, these results suggest that LC10a plays a similar role in the visuo-motor tracking of social targets across sexes.”) and added the following text in the conclusion to make this more explicit (Lines 279 – 281): “Our data on TuTuA activity in males during tracking of a fictive female stimulus combined with analysis of the connectome of the male optic lobe suggest that the same toggle switch operates across sexes.”

a. The most crucial case involves the gating of LC10a visual activity, which (to my knowledge) has not been directly observed in females. The authors record robust visual responses in a non-aggressive state in Figure 2a, which actually makes a strong case that the arousal-based gating seen in males is actually not present in females. Repeating the experiments of Figure 2a in a thermogenetically *alPg*-activated context would resolve this question. In the absence of this important experiment, the authors must clarify their reasoning, as this disrupts the logical chain from circuit mechanism to visual gating to behavioral context.

We thank the reviewer for raising this point. The LC10a work performed in Figure 3a (original Figure 2a) was done in males, rather than in females as stated by the reviewer. We apologize for not making this explicit. We have now added symbols throughout our figures to clarify when male or female flies were used, in addition to stating so in the figure legends and methods.

We believe that the data in Figure 3a and the published literature (Hindmarsh Sten et al., 2021, Ribeiro et al., 2018) is consistent with our working model that the gating of LC10a primarily occurs by regulating synaptic release at its axonal termini in the AOTu.

The goal of the recordings performed in Figure 3a was to examine the visual response properties of LC9 and LC10a neurons in the lobula as these were not assessed during a previous survey of the visual features detected by LCs (Klapoetke et al., 2022). Our LC10a data confirms previous work that found the dendrites of LC10a neurons in the lobula respond to fly sized objects under basal conditions (Ribeiro et al., 2018). Our

circuit model is that these responses are normally inhibited by TuTuA_2 neurons in the AOTu and only disinhibited when males initiate courtship or females initiate aggression. Therefore, these experiments serve as a confirmation rather than a contradiction and are consistent with the “logical chain from circuit mechanism to visual gating to behavioral context.” Importantly, these functional recordings were performed to compare the preferred object sizes of LC9 and LC10a with those experienced during male courtship and female aggression behavior (plotted in Figure 3d) and we do not draw conclusions beyond these statements.

b. The authors’ conclusion that selective modulation of LC10 subtypes shapes male courtship (Figure 5) is currently not well supported. The TuTu recording experiments, while elegant and informative, do not establish a causal role. And the behavioral activation experiments in Extended Data Figure 7g are missing some controls and could use some additional analysis. Metrics such as copulation success rate and time to copulation, which are often used to describe the mating effects of visually impairing males, would strengthen the conclusion. Further, the empty-gal4 negative control (see Tadres et al.), found in other experiments throughout the paper, is conspicuously lacking here.

We agree. We have rewritten this section to limit our conclusions to stating that the changes of opposite sign recorded in TuTuA_1 and TuTuA_2 in a male while tracking a ‘fictive female’ (moving dot) are consistent with the circuit model derived from the female aggression data. This is a much more direct and interpretable measure of visual gating – the focus of this manuscript – than courtship behavior, which is confounded by non-visual sensory input. For this reason (and because the individual who performed this work recently completed their PhD and moved to their postdoc in a different model organism), we have removed the data previously shown in the original Extended Data Figure 7g and corresponding text. We have altered the following text for clarification: Lines 261 – 263, “These results are consistent with our circuit model in females, supporting the notion that the same TuTuA-mediated switch may also be used in males to gate visual processing during social interactions.”

2. The functional connectivity experiments using Chrimson/GtACR need empty-gal4 controls. One control for each imaged/patched downstream neuron would suffice, to demonstrate that the observed responses are not due to nonspecific expression leak or a photoelectric/intrinsic photosensitivity effect. See Tadres et al. In the absence of these controls, I think the authors need to be careful about over-stating the functionality of connections suggested by their connectome analysis. I believe the control is especially critical for the dendritic disinhibition mechanism (mechanism 2). There are hints that nonspecific mechanisms might be relevant here, given the order of magnitude more stimulation required to minimally activate IB112 compared to TuTuAs.

We thank this reviewer for this comment and have performed the additional controls requested. We performed these experiments by expressing either UAS-Chrimson or LexAop-Chrimson without a GAL4 or LexA driver, respectively, as performed in Tadres et al., 2022. In these experiments, we found no nonspecific activation of IB112 or

TuTuA_1 or inactivation TuTuA_2 upon red light stimulation. Each of these controls were performed alongside a replication of the previous experiments, which further supports our functional connectivity data. For the IB112 experiments, we have also added the mecamylamine control along with additional calcium imaging experiments and controls to further support this data. We have updated the following three figures with these additional control experiments: Figure 5e – f, Extended Data Figure 5a – b, and Extended Data Figures 7e, j.

3. A core discussion topic is that the toggle switch mechanism described by the authors would act to better align information flow in the AOTu to the stimulus statistics of flies engaging in aggressive behavior. If my reading of the data is correct, the distribution of retinal object size in aggressive females and the tuning of LC10a are only glancingly aligned, with the peak of the retinal size distribution lying on the edge of the LC10a tuning curve. As such, the authors should soften their statements about LC10a's suitability for encoding visual features during aggressive bouts. Additionally, it's not clear what is known about the tuning of LC10c. It might make sense to perform another set of recording experiments (like the ones in Figure 2a) for LC10c. Then, the authors can show that aIPg activation shifts visual information flow from LC10c tuning (presumably even less well aligned with stimulus statistics) to LC10a tuning (as already described).

As noted above, since the data in Figure 3a (original Figure 2a) was performed in males, we have moved the data examining object size distribution during male courtship to the main figure to facilitate better comparisons between functional and behavioral tuning (Figure 3d). From this data, we can see that the peak of the tuning curve for LC10a aligns with the object sizes during male courtship and that those during female aggression are a component of this. We should note that the female aggression classifier captures interactions where females are closer to each other, so it is expected that the distribution is not as broad as in male courtship.

In terms of LC10c recordings, unfortunately, we have yet to identify a genetic driver line labeling only LC10c, despite extensive multi-year efforts in our lab and others to obtain such a line. We have added additional connectomic analysis and corresponding text to illustrate LC10a and LC10c receive largely non-overlapping inputs in the optic lobe, suggesting that LC10c might be tuned to chromatic cues (Figure 3b – c; Lines 119 – 124: “We found that while the inputs to either LC10a or LC10c do not differ across sexes, inputs to these two cell types show little overlap indicating that LC10a and LC10c respond to distinct visual features (Figure 3b – c). For example, LC10c receives approximately 30% of its inputs from interneurons directly downstream of the R7 (Tm5a and Tm5b) and R8 (Tm20) photoreceptors that are critical for color vision, whereas LC10a receives less than 4% of its input from such interneurons.”) Additionally, these data support our speculation that most of the circuit we describe does not differ across sexes as these inputs are similar in males and females.

Beyond these concerns, I have a few data presentation/statistics issues. First, I found the presentation method for genotypes/genetic controls quite difficult to interpret. I

believe a general revision of bar plot labeling is called for here. A reader would not quickly grok what is actually being tested in Fig 2c, for example. Across figures, “positive/negative control” can mean different things. Filling in a bubble with “aIPg active” in an experiment where GtACR silencing occurs is misleading. And so on. Rather than using these simple word models for each experiment, I believe the standard approach of showing which alleles and/or manipulations are present in each case would be much clearer, and there would be less risk of misinterpretation.

We agree with the reviewer and have re-organized each of these figures for clarity. Figures 3f, 4c, 6a – b and Extended Data Figures 1b, 3g, and 8d have all been re-organized to include the genotype in the figure.

Finally, there is a good deal of variance in the group sizes used in a number of statistical tests. Most worrying are the TuTuA experiments in Figure 5a, with group sizes ranging from 6 to 20. Given the trends towards effects in the empty-gal4 controls that can be clearly seen in Extended Data Figure 7b&c, dramatically different group sizes are problematic. I believe the authors should collect additional data to bring the group sizes into better alignment, so that readers can be more confident in the paper’s conclusions.

In response to the reviewer’s comment, we have repeated both the control and experimental groups to substantially increase the group sizes and make them more uniform, as the reviewer suggested, with the group sizes being compared as follows: Figures 3f (n = 23 – 26), 4c (n = 17 – 22), 6a – b (n = 19 – 24) and Extended Data Figure 3g (n = 15 – 28). These additional experiments further support our previous results. The only change to note is that in Figure 6b, the empty stable split line (negative control), exhibited a slight decrease in aggressive behavior during the stimulus period. As this decrease is opposite to the effect observed in our experimental group (TuTuA_2 > Chrimson), this trend did not change the conclusions of this experiment. We noted this in the figure legend.

Minor Comments:

1. The authors refer to the sensory gating here as “visual attention” a few times, and I’m not confident this term applies. “Visual attention” typically refers to gain enhancements in a spatial subset of the visual field, whereas this paper describes gain enhancements to particular visual features, anywhere in the visual field. I believe “context-dependent gating of visual features” is more appropriate than “visual attention.”

We agree with the reviewer that there are multiple definitions of visual attention. We have added the following in the text for clarification (Lines 33 – 35; “Such focus or ‘feature-based attention’¹ also occurs during certain behavioral states, including aggression when it is important to be attuned to the movement of a competitor.”). Additionally, we now cite Carrasco, 2011 review on visual attention (Citation #1), which states that feature-based attention is one of three main types of attention and “guides an observer to particular features in the visual scene.”

2. I believe a citation of Turner et al. is more than warranted, as it is one of the few prior examples of LC modulation by behavioral context.

We agree and thank the reviewer for this citation. We have added it to the current version of the manuscript (Citation #32).

3. In Figure 2 and Extended Data Figure 2, there is a clear effect on aggression mediated by LC10b/c neurons. However, the particular way the data are summarized obscures this effect in the main Figure. While the author's wording is technically correct ("LC10a, but not LC10b/c, resulted in a sustained decrease..." [emphasis is mine]), I don't think we are doing justice to the complexity of the data by pushing this aside. The sign of the effect mediated by LC10b/c seems like it might be consistent with the 10a vs. 10c toggle switch mechanism. Could this result be folded into the narrative?

While we agree with the reviewer that the off kinetics for LC10bc inactivation appear to be consistent with the toggle switch mechanism, this conclusion cannot be made as the empty stable split also exhibits a similar transient decrease in aggression in the stimulus off period, likely reflecting non-specific visual effects. We have now plotted the LC10bc and empty-split experiments on the same graph in Extended Data Figure 3c to facilitate direct comparisons.

4. In Figure 4c-d, I am having a hard time assessing the degree to which the TuTuA inputs are axo-axonic. The synapse sites don't appear particularly close to the yellow presynaptic terminals. Is this because we are only looking at a portion of the larger arbor? If so, having a full skeleton present that also shows postsynaptic sites, and then doing the existing images as zoomed-in insets, might help to clarify that these are indeed close to specifically presynaptic sites.

We appreciate the reviewer for this point. For clarification, we have now added zoomed in inset images of the example LC10a and LC10c neurons as suggested and scale bars to Figures 5c – d.

5. In Figure 4e-f, a non-nAChR-mediated residual is present, but I was unable to find a discussion of this in the text. What do you think is the source of this input? Is this a problem for your model, and if not, why?

We agree with the reviewer that there is a slight residual current present, which is also not present in the Chrimson controls. However, we have no way to distinguish between possible sources of this small residual signal, which may include incomplete action of the inhibitor in this cell type, the presence of additional neurotransmitters and neuropeptides, residual postsynaptic potential via non-cholinergic transmitter release, or a polysynaptic pathway. For this reason, we have chosen not to speculate on the source of this remaining activity.

6. In Figure 5b, adding SMP054 and LC10a labels to the schematic will help improve clarity and consistency.

We have added this to Figures 6b (original Figure 5b).

7. The scale of the Y-axis in Figure 5b is dramatically different from every other plot of this kind in the manuscript. I know this must be the case in order to properly illustrate the effect, but I wonder if there is something that can be done to make the plot a little more visually different from the others, to cue readers into this fact at a glance? Perhaps not having it horizontally aligned with the plots in panel a would be enough.

We thank the reviewer for this comment and have adjusted the Y-axis on Figure 6b (original Figure 5b) to introduce a break in the scale to visually indicate it is different from other plots.

8. In Figure 6, I believe the toggle-switch mechanism would be easier to appreciate if you split the LC10 population into LC10a and LC10c. Then this would be a great way to see all three mechanisms in the same circuit diagram, and how they relate to the proposed output pathway in the AOTu.

We have done this in Figure 1b and Extended Data Figure 10.

9. Lines 121-122: “limited subset of LC neurons tuned to fly-sized moving objects” – my understanding from Klapoetke et al. and Turner et al. is that a large fraction of recorded LC/LPLC types respond to small moving objects. And as the authors explain, with varying distance, the size of a “fly-sized object” will change. As such, I think the “limited subset” phrase should be removed. Additionally, I think it would be fair to add some graphics or language about the aIPg connectivity to other moving spot-tuned cell types, here and/or in Extended Data Figure 9.

This phrasing has been removed (see Line 124). We also include connectivity information to synaptic targets of other LCs that detect small object in the text (Line 97 – 100; “Only three other downstream neurons receive significant input from both LC and aIPg neurons: PVLP120 receives 30% of its synaptic inputs from LC17, 19% from LC12 and 1.5% from aIPg; SMP312 receives 5.3% of its inputs from LC21 and 4.6% from aIPg; and PVLP006 receives 35% of its inputs from LC6, 11% from LC16 and 2.3% from aIPg.”).

10/11. Lines 207-208: unless I’m missing something in the circuit diagram, the idea that this organization would primarily mediate feedback inhibition seems a bit weak, given the axo-axonic connectivity pattern.

Lines 208-211: this sentence is a bit opaque, it’s not clear what interrogations are being proposed, nor is it clear why they can’t currently be done. Does this point need to be made?

These sentences identified by the reviewer are meant to highlight the alternative hypotheses and caveats of the conclusions we made from our connectomic analyses. We have made this more explicit by changing the first sentence of this paragraph to: (Lines 211 – 213) “We recognize that our interpretation of the mechanisms suggested by the circuit diagram are most likely incomplete, and there are features of this circuit that cannot be fully understood from the connectome alone.”

12. Line 249: I'm not sure it's fair to say that baseline physiological properties of TuTu neurons are the same across sexes based only on the data shown in Extended Data Figure 8. The action potential shape is shown to be different, and there seems to be significant differences in burstiness/rhythmicity that may signal differential expression of ion channels. Without more analyses and/or experiments, I recommend softening this language.

We agree with the reviewer and have changed this wording in the text to (Line 254 – 255) “...share aspects of their baseline physiological properties...”

13. Line 625: I believe there is a typo in a number of genotypes written in legends. This is one example. Switching to the conventional genotype-based bar plot labels (see above) will obviate the need for these big genotype lists.

We have included the genotypes in Figures 3f, 4c, 6a – b and Extended Data Figures 1b, 3g, and 8d as suggested and therefore removed this from most of the figure legends.

14. Line 720: Should add “optogenetic” here (in reference to the stimulus). Related, why is the activation so long here compared to the patching experiments? Related to major concern 2, above.

Optogenetic has been added to the figure legend for Extended Data Figure 5b, and 7a – e.

The difference in the stimulation protocol is due to the fact that we are probing different timescale responses in the two experiments. For whole cell patching, we aimed to confirm direct synaptic connectivity and so wished to elicit time-locked action potentials in the pre-synaptic neurons, while stimulation during calcium imaging was meant to examine the neuronal activity over a similar timescale as the behavior.

References:

Hindmarsh Sten T, Li R, Otopalik A, Ruta V. Nature 2021

Jovanic T, Schneider-Mizell CM, Shao M, Masson JB, Denisov G, Fetter RD, Mensh BD, Truman JW, Cardona A, Zlatic M. Cell 2016

Klapoetke NC, Nern A, Rogers EM, Rubin GM, Reiser MB, Card GM. Neuron 2022

Tadres D, Shiozaki HM, Tastekin I, Stern DL, Louis M. bioRxiv 2022

Turner MH, Krieger A, Pang MM, Clandinin TR. Elife 2022

Referee #3 (Remarks to the Author):

The circuits and mechanisms of early visual processing is rather well understood across various animal clades, particularly in insects (or arthropods generally) and vertebrates. However, while a wealth of studies in mammals (particularly based on electrophysiological approaches) has identified strikingly complex concepts of visual representations in deeper brain regions and to some degree the circuit organization the corresponding counterparts are less well understood in non-vertebrates.

The current manuscript provides an elegant and powerful example of visual processing of potentially complex type of cues using the fruit fly as model. It is a beautiful example of how connectomics may be employed together with high-precision genetic tools and physiological and behavioral analyses to decipher how higher brain circuits are able to achieve surprisingly advanced functions. It hereby focusses on the state-dependent change in visual processing in female flies.

There are a series of intriguing and highly relevant points in the study. Above all, it shows that an internal state such as aggression can alter how visual information is processed; I find this particularly relevant since aggression is by itself a rather complex state (as compared to starvation). Secondly, the authors layout three different state-dependent circuit motifs in a rather great vigor.

I do not have any major criticism at the work, however there are a few points, that may benefit from clarification or further explanation.

While connectomic data are powerful to depict (and to some degree proof) the existence of chemical synapses between certain neurons, there are several limitations to the analyses. One particularly relevant limitation to the current work is that the connections can merely be defined by the highest or most abundant synapses, however the actual synaptic weight is much more critical than synapse number (e.g. 10 strong synapses may be overcoming 100 weak ones). Do the authors have any evidence in this specific circuit for a correlation of this?

The reviewer highlights a general limitation in interpreting connectomic information. While we only discuss connections with a high number and percentage of synapses, we agree with the reviewer that we cannot determine from the connectome the physiological strength of these synapses. To our knowledge, no one has yet

experimentally assessed the degree to which the number of synaptic contacts between neurons correlates with the magnitude of the post-synaptic potential.

The physiology in this paper confirms functional connections, but their relative “strength” cannot be determined by the experiments here. Therefore, we have replaced all references in the text to “strong connections” to “highly” or “most” connected.

Similarly, while they show axonal termini, do they have evidence that the actual synapse number in an average female fly is indeed similar to the one in the connectome (which represents a specific snapshot of one individual)?

Synapse number generally does not vary widely between individuals in *Drosophila*, at least in cases of neurons connected by more than 10 synapses (Schlegel et al., 2023). All the connections examined consist over 50 synapses. Schlegel et al., 2023 found no cases in the entire connectome of such abundant connections differing significantly across individuals.

Do the authors have any evidence for the molecular input pathways of the “aggression state”? The behavioral paradigm is only briefly mentioned and since it’s a core aspect of the manuscript may deserve better explanation. As it reads now it’s rather a miraculous state-switch, without much insight into the cues and input pathways.

We thank the reviewer for this interesting point. Our current work focuses on describing how, once aIPg is activated, it modifies downstream circuits to alter behavior. For this reason, nearly all of our experiments involve exogenous activation of aIPg, providing tight control over the aggression state, but do not speak to the important question of what excites aIPg to initiate this state. Our previous work demonstrated that aIPg stimulation triggers a persistent aggressive internal state (Chiu et al., 2023). Therefore, we discuss aIPg activation in terms of an aggressive state. We previously mapped the inputs to aIPg (Schretter et al., 2020), which included pC1d and pC1e, neurons that also promote aggression. We also included whole-cell patch clamp recordings in Extended Data Figure 7g – k demonstrating that pC1d and pC1e are also connected to TuTuA_2.

In terms of detailing the behavior, we agree with the reviewer. We have now added additional description in the text (Lines 142 – 146; “To assess the involvement of specific cell types in aIPg-mediated aggression, we recorded, tracked, classified, and analyzed the aggressive behavior of groups of female flies in an arena as diagrammed in Figure 3e. We quantified the average percent of trajectories in which flies were performing aggression in each such experiment, which we represented as a single point in our graphs and analyzed across approximately 20 experiments per condition (Figure 3e).”) and a diagram in Figure 3e. The trajectory for a single fly is occasionally discontinuous throughout the course of a video due to tracking difficulties. We have clarified this in the methods (Lines 891 – 896).

While the connectome is based on a female individual and much of the current work is focused on female aggression, the authors also provide evidence for the presence of

the same circuit in male (courtship). This is particularly intriguing, since the recognition of a conspecific fly (independent of male/female) elicits different responses in both sexes. Is it to some degree or approximation possible to explain this in the current knowledge of male/female specific circuits and the existence of split *gal4* lines and sex-specific neural lines?

We agree with the reviewer that this is an intriguing point. In this work, we focused on a shared component feature of aggression and courtship (visual tracking). Our hypothesis is that cell types which differ across sexes (*aIPg*, *pC1/P1*) regulate common nodes (i.e. *LC10s*, *TuTuAs*, *IB112*) to generate visual tracking of social targets across sexes. The framework of layering connectomics analysis with genetic tools, behavioral analysis, and functional imaging that we laid out in this work could also be used to explore sex variable features of a behavior (i.e. avoidance vs approach towards conspecific following recognition). However, more detailed analysis will be required once the male connectome is available.

Reviewer Reports on the First Revision:

Referees' comments:

Referee #1 (Remarks to the Author):

In general, the authors have addressed most points raised by us and the other reviewers. However, there are few remaining issues which should be corrected before the ms is published.

1. The authors did not correct LC10 to LC10s or LC10 types in most instances it was used. Importantly, Figure 1b and Extended Figure 10 still have LC10 and not the LC10 types. Reviewer #2 also requested this change in these figures, and the authors reply they did this, but they didn't.

Line 68: LC10 should be LC10s;

Line 92: LC10 neurons should be LC10-group neuron types;

Line 174: LC10 neurons should be LC10-group neuron types or LC10s;

Line 188: LC10 shows up again alone.

2. The authors' response to our previous point 7: The papers the authors cite concern female aggression only. While male courtship may be context dependent - sensory context and experience - it is highly stereotypical as long as the male is socially isolated and naïve, the female is virgin, and the circadian rhythm activity peak is respected – the base level is consistently high, > 80% of time spent courting. It might be possible to play with context – either using mated females, place food in the arena, or both, etc – in which the baseline levels of female-female aggression are higher and therefore usable to test whether the circuits mechanisms here proposed are in action in 'naturalistic' aggression or baseline aggression. Did the authors ever try to increase this baseline? For instance, using a mated and unreceptive female rejecting a male? In their rebuttal letter, the authors emphasize that they are testing circuits underlying aIPg mediated aggression. This should also be clearly pointed out in the main text.

3. Line 153 – 154 “ Collectively, these results...” Here the authors combine the object size tuning done in males (Figure 3a) with aggression behavior done in females (Figure 3f). This sentence needs to include references.

4. Line 239: The authors might want to specify that the toggle switch in females might really be an ON or OFF switch, and not necessarily a modulation of gain, as the text currently implies (see also comment from Reviewer #2).

5. Line 241 to 247: There is barely any information about LC10 neuron types other than LC10a and LC10c. The connectomic data in figure 2a does show connections of LC10a, b, c and d. But there is no evidence that aIPg will likely gate all visual information coming through the AOTu. First there are more LC10-types found by Sebastian Seung (LC10e, LC10f). Second, LC10b and LC10d downstream neurons are not well characterized. The sentence in line 145 “Thus, aIPg is primed to regulate the flow of information through all visual AOTu interneurons...” is an overstatement. It would help to break down 'LC10' into LC10 types in Figure 1b and Extended Data Figure 10.

6. Line 166: The authors probably mean Extended Data Figure 5c – e and not 4c – e.

Referee #2 (Remarks to the Author):

This revised manuscript has been considerably strengthened relative to the original submission. Many thanks to the authors for their careful and diligent work in response to the reviewers' comments and questions. The additional behavioral experiments and negative controls are especially appreciated, and the figure updates make the work much easier to understand.

I have one lingering concern, and I want to apologize to the authors for doing a poor job explaining my position in the original review. I hope that I can be a bit clearer now with a second look at the data.

Much of the significance of this study (as set up in the abstract, intro, and even the paper's title) hinges on the presumption that LC10a and LC10c handle distinct sensory features, and the implication that LC10a is better suited to encoding fly-sized visual features during aggressive bouts. While the authors do not make these claims directly, they are heavily implied by the framing of the manuscript around "visual feature attention" and "behavioral state gating." Given that the authors are not currently able to record the selectivity of LC10c for comparison, and the lingering issue that LC10a tuning only overlaps a portion of the retinal size distribution during aggression, I would still like to see the authors soften this line of reasoning. They show that LC10c receives different input than LC10a, but we do not know the size tuning of the different inputs in question (Tm5s, Tm20). As such, we cannot conclude from this data alone that LC10a is better suited to visual encoding during aggression.

The authors have also not done in females the experiment that demonstrated state-specific gating of visual responses in LC10a neurons (Hindmarsh Sten 2021). This limits our ability to infer circuit function similarity across sexes (although I heartily agree that the circuit architecture is similar). In other words, this paper demonstrates a circuit mechanism that *could* gate LC10 transmission, but that gating itself has not been shown in this context. The read-outs of their circuit dissection experiments look at behavior or the outputs of TuTuAs, not at the outputs of LC10a and LC10c. Thus, it is clear that aPG exerts control over aggressive behaviors by the circuit mechanisms described. But I do not think we can be certain that this control represents "visual feature attention" or "behavioral state gating" of visual information flow.

I believe text adjustments alone are enough to address this concern, including changes to the paper title and abstract that should not lean so heavily on the idea that visual information flow is altered during aggressive behavior.

Two smaller points:

(1) The title of Fig 5 needs to be updated in accordance with the authors' new position that selective modulation of LC10 subtypes is not necessarily involved in male courtship (as discussed in my original review).

(2) Some of the figure pointers in the extended data figures do not reflect the updated figure numbering.

Referee #3 (Remarks to the Author):

In the revised manuscript the authors have added several helpful and critical clarification as well as minor experiments to support their model. I do not have any further comments and congratulate the authors to this intriguing piece of work.

Author Rebuttals to First Revision:

Schretter et al., 2024
Response to Reviewers

We would like to thank the reviewers for their further comments. A point-by-point response is provided below.

Referees' comments:

Referee #1 (Remarks to the Author):

In general, the authors have addressed most points raised by us and the other reviewers. However, there are few remaining issues which should be corrected before the ms is published.

1. The authors did not correct LC10 to LC10s or LC10 types in most instances it was used. Importantly, Figure 1b and Extended Figure 10 still have LC10 and not the LC10 types. Reviewer #2 also requested this change in these figures, and the authors reply they did this, but they didn't.

We have now changed this from LC10s to LC10-group types in both Figure 1b and Extended Data Figure 10.

Line 68: LC10 should be LC10s;
Line 92: LC10 neurons should be LC10-group neuron types;
Line 174: LC10 neurons should be LC10-group neuron types or LC10s;
Line 188: LC10 shows up again alone.

We have changed these from LC10 to LC10-group cell types.

2. The authors' response to our previous point 7: The papers the authors cite concern female aggression only. While male courtship may be context dependent - sensory context and experience - it is highly stereotypical as long as the male is socially isolated and naïve, the female is virgin, and the circadian rhythm activity peak is respected – the base level is consistently high, > 80% of time spent courting. It might be possible to play with context – either using mated females, place food in the arena, or both, etc – in which the baseline levels of female-female aggression are higher and therefore usable to test whether the circuits mechanisms here proposed are in action in 'naturalistic' aggression or baseline aggression. Did the authors ever try to increase this baseline? For instance, using a mated and unreceptive female rejecting a male? In their rebuttal letter, the authors emphasize that they are testing circuits underlying aIPg mediated aggression. This should also be clearly pointed out in the main text.

As stated in the methods, all experiments are done in mated females. We also tested "naturalistic" aggression (that is not induced by aIPg activation) when we look at the role of the TuTuAs in Figure 6b.

In response to the reviewers suggestion, we added the following sentence to lines 71 – 73: “Here, we focus on female-female aggression and show that aIPg uses three circuit mechanisms to modulate visual processing, which underlies this social behavior (Figure 1).”

3. Line 153 – 154 “Collectively, these results...” Here the authors combine the object size tuning done in males (Figure 3a) with aggression behavior done in females (Figure 3f). This sentence needs to include references.

We cited relevant references in the previous paragraphs when the individual points were first raised.

4. Line 239: The authors might want to specify that the toggle switch in females might really be an ON or OFF switch, and not necessarily a modulation of gain, as the text currently implies (see also comment from Reviewer #2).

We added this point to Line 224 – 225: “In the simplest interpretation of the circuit, TuTuA_1 and TuTuA_2 would function as a reciprocal ON/OFF switch for LC10a and LC10c.”

5. Line 241 to 247: There is barely any information about LC10 neuron types other than LC10a and LC10c. The connectomic data in figure 2a does show connections of LC10a, b, c and d. But there is no evidence that aIPg will likely gate all visual information coming through the AOTu. First there are more LC10-types found by Sebastian Seung (LC10e, LC10f). Second, LC10b and LC10d downstream neurons are not well characterized. The sentence in line 145 “Thus, aIPg is primed to regulate the flow of information through all visual AOTu interneurons...” is an overstatement. It would help to break down ‘LC10’ into LC10 types in Figure 1b and Extended Data Figure 10.

On the specific point about aIPg regulation, all AOTu neurons that get input from LC10s get input from LC10a and LC10c (see Figure 2a). Thus, the toggle switch that only affects LC10a and LC10c would still affect “the flow of information through all visual AOTu interneurons...” To clarify this point, we replaced “by gating which LC10s can effectively signal to them” with “by gating whether LC10a or LC10c can effectively signal to them” in Line 259 – 260.

On the point of the other LC10s, we focus on LC10a and LC10c because these are the two LC cell types that appear to be strongly regulated by aIPg. We added the following sentence to the legend of Figure 5: “TuTuAs connect to specific LC10s: 98% of TuTuA_1’s synapses onto LC10s go to LC10c and 98% of TuTuA_2’s go to LC10a. TuTuA_1 and TuTuA_2 make <1% of their synapses onto LC10b, LC10d, LC10e or LC10f.”

We also modified the legend to the Extended Data Figure 10 to include these cell types, adding: “On average, the number of Li22 inputs to individual cells in each of the LC10-group cell types are as follows: LC10a, 23; LC10b, 12; LC10c, 27; LC10d, 32; LC10e, 4; and LoVP76/LC10f, 0 in males and LC10a, 8; LC10b, 6; LC10c, 14; LC10d, 15; LC10e, 1.4 and LC10f 1.5 in females.”

6. Line 166: The authors probably mean Extended Data Figure 5c – e and not 4c – e.

We have changed this in the updated text.

Referee #2 (Remarks to the Author):

This revised manuscript has been considerably strengthened relative to the original submission. Many thanks to the authors for their careful and diligent work in response to the reviewers' comments and questions. The additional behavioral experiments and negative controls are especially appreciated, and the figure updates make the work much easier to understand.

I have one lingering concern, and I want to apologize to the authors for doing a poor job explaining my position in the original review. I hope that I can be a bit clearer now with a second look at the data.

Much of the significance of this study (as set up in the abstract, intro, and even the paper's title) hinges on the presumption that LC10a and LC10c handle distinct sensory features, and the implication that LC10a is better suited to encoding fly-sized visual features during aggressive bouts. While the authors do not make these claims directly, they are heavily implied by the framing of the manuscript around "visual feature attention" and "behavioral state gating." Given that the authors are not currently able to record the selectivity of LC10c for comparison, and the lingering issue that LC10a tuning only overlaps a portion of the retinal size distribution during aggression, I would still like to see the authors soften this line of reasoning. They show that LC10c receives different input than LC10a, but we do not know the size tuning of the different inputs in question (Tm5s, Tm20). As such, we cannot conclude from this data alone that LC10a is better suited to visual encoding during aggression.

Our statement is based not of these data alone, but also the behavioral data showing that LC10a, but not LC10bc, plays an important role in aggressive behavior during aP_g activation. We believe this indicates that LC10a is "better suited" to visual encoding during aggression.

The authors have also not done in females the experiment that demonstrated state-specific gating of visual responses in LC10a neurons (Hindmarsh Sten 2021). This limits our ability to infer circuit function similarity across sexes (although I heartily agree that the circuit architecture is similar). In other words, this paper demonstrates a circuit mechanism that *could* gate LC10 transmission, but that gating itself has not been shown in this context. The read-outs of their circuit dissection experiments look at behavior or the outputs of TuTuAs, not at the outputs of LC10a and LC10c. Thus, it is clear that aP_g exerts control over aggressive behaviors by the circuit mechanisms described. But I do not think we can be certain that this control represents "visual feature attention" or "behavioral state gating" of visual information flow.

The reviewer states that: "it is clear that aP_g exerts control over aggressive behaviors by the circuit mechanisms described." Given this, we do not think that physiological measurement of LC10a transmission is needed to support the statements made in the manuscript. More specifically, as all circuits mechanisms we describe involve neurons that participate in visual

information flow, it seems appropriate to conclude that visual information flow is what is modified. It is also clear from our behavioral data that inactivating LC10a decreases orientation to neighboring flies during aIPg stimulation, while the inactivation of LC10c (together with LC10b) does not. Females engage in the visual pursuit of another fly when they are in an aggressive state indicating that the response of an aggressive female to fly-sized visual objects has been altered. For these reasons, we believe that use of “visual feature attention” or “behavioral state gating” is appropriate for this context.

I believe text adjustments alone are enough to address this concern, including changes to the paper title and abstract that should not lean so heavily on the idea that visual information flow is altered during aggressive behavior.

We believe that the wording of the abstract is already tempered. However, in response to this reviewer’s concern, we did replace “selectively amplify” with “modify” in Line 19 of the abstract and modified the title to change the term from “gates” to “alters,” a less mechanistically loaded term. The title now reads: “Social state alters vision using three circuit mechanisms in *Drosophila*.”

Two smaller points:

(1) The title of Fig 5 needs to be updated in accordance with the authors’ new position that selective modulation of LC10 subtypes is not necessarily involved in male courtship (as discussed in my original review).

In Figure 6 (originally Figure 5), we replaced “LC10s” by “TuTuA_1 and TuTuA_2.” We measured the activity of these cell types in males during fictive courtship pursuit.

(2) Some of the figure pointers in the extended data figures do not reflect the updated figure numbering.

We updated this in the text.

Referee #3 (Remarks to the Author):

In the revised manuscript the authors have added several helpful and critical clarification as well as minor experiments to support their model. I do not have any further comments and congratulate the authors to this intriguing piece of work.